# Seeing What's Not There: Negation Understanding Needs More Than Training

**Bhuvan Aggarwal**  **Amit More**  **Mudit Soni**  **S Divakar Bhat**

**Honda R&D, Tokyo**

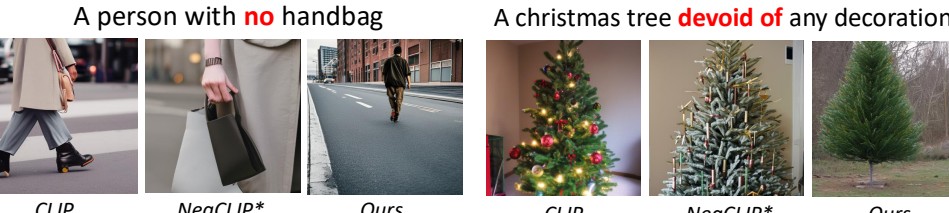

A person with **no** handbag

*CLIP*  *NegCLIP\**  *Ours*

A christmas tree **devoid of** any decoration

*CLIP*  *NegCLIP\**  *Ours*

Figure 1: **Vision-Language Models struggle with negation understanding.** Standard CLIP-based Text-to-Image generation models fail to understand the negation in input text, generating images containing the negated objects. Even when models are fine-tuned on exclusive datasets with hard-negatives for negation understanding, they struggle to understand negations (NegCLIP*). Our proposed embedding correction approach on CLIP text embeddings successfully addresses the negation understanding failures, generating images which exclude the negated objects.

## Abstract

Understanding the negation in a sentence is an important part of compositional understanding and logic in natural language. Many practical AI applications, such as autonomous driving, include precise instruction with negations. For example, following instruction to an AI assistant "locate a parking spot without a vehicle" requires the assistant to not confuse between presence and absence of vehicles. Although joint embedding-based Vision Language Models (VLMs) like CLIP have revolutionized multi-modal tasks, they struggle to interpret negation. To address this limitation, recently many works proposed to solve the problem through a data-centric approach by introducing additional datasets with hard-negative samples for both image and text data. Contrary to these approaches, we present a zero-shot approach to tackle the negation understanding problem. We probe the properties of CLIP text embeddings and show that they follow compositional arithmetic operations, which allow the addition or removal of semantic information directly in the embedding space. We then present a rule-based approach to extract negated text from given caption and then use it to explicitly remove corresponding semantic information from original embedding, improving negation understanding in VLMs. Our approach does not require expensive training process to induce negation understanding into the model, and achieves the state-of-the-art performance on popular benchmark for negation understanding. We improve baseline CLIP model performance on NegBench from 25.5% to 67.0% for MCQ and from 50.9% to 56.1% for retrieval tasks. Even NegCLIP model which is fine-tuned on negation datasets, our approach boosts its MCQ accuracy from 54.03% to 66.22% and retrieval accuracy from 59.25% to 60.1% showing strong performance.

## 1 Introduction

Vision-Language models (VLMs) have demonstrated remarkable capabilities in multi-modal tasks. This is generally achieved using a contrastive learning framework by aligning image and text information in a joint-embedding space, such as in CLIP models Radford et al. (2021); Li et al. (2022b).

These models are widely used for tasks such as text-to-image generation Choi et al. (2022); Rombach et al. (2022); Ramesh et al. (2021), cross-modal retrieval Chen et al. (2023); Luo et al. (2022), image captioning Vinyals et al. (2015); Xu et al. (2015) and referring image segmentation Wang et al. (2022); Lee et al. (2023); Lüddecke & Ecker (2022).

However, it has been shown that these models generally struggle with compositional understanding of text Fan et al. (2024); Yuksekgonul et al. (2022); Hsieh et al. (2023); Ma et al. (2023); Lewis et al. (2022) and, in particular, negation Singh et al. (2024); Li et al. (2025), a core linguistic aspect of understanding and reasoning. In general, the CLIP models often ignore the negation words such as "no", "not" and "without", referred to as the "Affirmation Bias" Li et al. (2025). Negation is essential for precise semantic interpretation in natural language with many practical applications. For example, following text prompt to a diffusion model, "a photo of a car with tires" and "a photo of a car without tires" will result in similar generated images, both with tires, which is very detrimental for their performance and generalization.

The failure to induce negation exposes the shortcoming in how these models structure and interpret semantic concepts in the embedding space. This limitation of joint-embedding models can be attributed to both the training data and process. The CLIP model is trained on web-scale image-text pairs using contrastive learning. The negation data is highly underrepresented in these pairs as shown in Yoon et al. (2025), with Laion-400M dataset Schuhmann et al. (2021) containing less than 1% negated captions. Even when present, the negated captions don't align with corresponding image in terms of the negation context. This problem is further aggravated by the contrastive learning approach which encourages the model to interpret the text as a bag-of-words rather than understanding its compositionality Yuksekgonul et al. (2022). In response to this, most of the previous research has tried to tackle this problem using a data-centric approach by introducing new datasets with explicit hard-negatives to improve model learning. Though effective, these approaches still struggle to understand negations completely and solely relying on large-scale datasets for performance improvement is not an optimal solution.

In the present work, we focus on an approach orthogonal to data-centric methods and provide an alternative solution to improve negation understanding. We ask an important question "*Do we really need fine-tuning on special datasets with hard negatives to improve negation understanding in VLMs?*". We show that a simple post-hoc correction to the text embeddings of CLIP (not fine-tuned on such datasets) can outperform even the fine-tuned variants by a large margin. In particular, we propose to solve negation understanding problem in a *novel zero-shot, training-free* approach through explicitly *enforcing compositionality constraints* on the text embeddings to reflect the effect of negation in the embedding space. We note that negation manifests itself as a directional offset in embedding space Li et al. (2025). We first estimate this directional offset as a correction signal for negation understanding and then construct a negation-aware embedding vector.

Our approach is inspired by the linearity shown by the word vector models and vision-language embeddings Mikolov et al. (2013); Couairon et al. (2022) where semantic information can be manipulated using basic arithmetic operations such as "King" - "Man" + "Woman" = "Queen". The idea that CLIP embeddings can be manipulated directly via vector arithmetic is not new, and is discussed in detail in Section 2.3. Our approach is primarily motivated by an empirical and geometric observation. We test our approach on various benchmarks and show that our zero-shot approach significantly boosts the negation-understanding of CLIP models.

The key contributions of our work are summarized as follows:

- We identify and characterize the limitation of existing models in terms of negation understanding. We show that CLIP text embeddings can be corrected using compositional arithmetic operations. Our approach computes the correction signal using directional offset and generates final negation-aware embedding for better negation understanding.

- We propose a training-free zero-shot embedding correction approach to improve CLIP's ability to interpret negation. Our approach achieves best results using a baseline CLIP model—not trained on explicitly curated negation datasets—outperforming best models in the literature which are trained on such hard-negative datasets.

- We show strong limitations of existing data driven approaches which simply curate millions of image-text pairs and hope to solve negation understanding by existing contrastive learning framework. Simplicity of our method—training free approach and strong perfor-

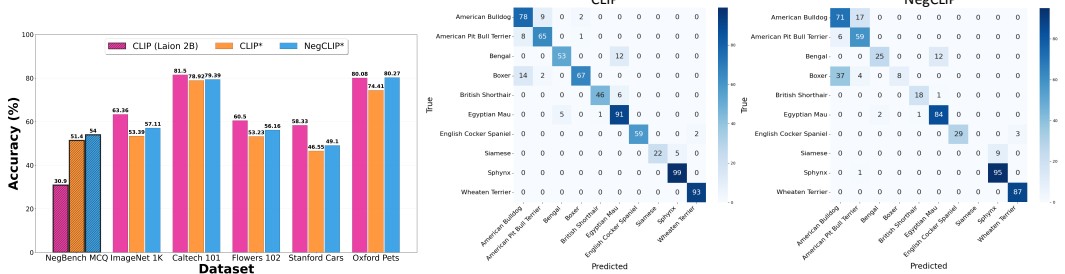

Figure 2: (1) We compare zero-shot image classification accuracies across multiple datasets for different CLIP models. * indicates models fine-tuned on the CC12M-NegFull dataset from Li et al. (2025) improving score on NegBench-MCQ benchmark (shown on left) but also leading to reduced foundational knowledge and decrease in zero-shot image classification accuracies. (2) Confusion matrices on Oxford Pets dataset shows that NegCLIP* have increased confusion between categories, demonstrating how negation-focused fine-tuning can degrade general capabilities of model.

mance—suggests that real improvement to negation understanding may need a new training framework to effectively utilize given data.

## 2 RELATED WORK

### 2.1 LIMITATIONS OF VISION-LANGUAGE MODELS

Early investigations into CLIP model's capabilities include the ARO Yuksekgonul et al. (2022) and CREPE Ma et al. (2023) benchmarks. ARO aims at evaluating the VLMs' ability to understand relationships, attributes and order and proposes NegCLIP while CREPE presents a benchmark to evaluate VLMs on two important aspects of compositionality: systematicity and productivity. Hsieh et al. (2023) show that these benchmarks are biased and proposes an unbiased dataset SugarCREPE, using LLM-generated hard negatives for fluent and meaningful captions. Further, Hsieh et al. (2024) proposes Graph-based captioning to enhance the training data using explicit compositional structure.

### 2.2 ADDRESSING NEGATION

ConCLIP Singh et al. (2024) introduced CC-Neg dataset with hard-negatives, built on the CC-3M dataset Sharma et al. (2018). However, CC-Neg contains hard negatives only as a distraction and not as a target ground truth. Following this, many works Li et al. (2025); Yoon et al. (2025) have introduced better benchmarks, datasets, and fine-tuned CLIP variants. Li et al. (2025) creates large-scale synthetic datasets-C12M-NegFull, based on CC-12M Changpinyo et al. (2021) and the NegBench benchmark for evaluation. Similarly, TripletCLIP Patel et al. (2024a) introduces hard negative samples in visual domain. General trend here, in research community, is to improve the quality of training datasets with minor modifications to contrastive loss by adding distractor samples or hard negatives. However, relying on retraining exposes models to catastrophic forgetting of the foundational knowledge, affecting the generalization performance on unseen domains or negation contexts.

### 2.3 TEXT EMBEDDING EDITING AND ZERO-SHOT COMPOSITION

Orthogonal to the data-centric approach, our method explores manipulating the text embeddings directly. The embedding arithmetic method has been shown to be effective in the traditional NLP literature, also known as the "Vector-offset method" Mikolov et al. (2013), often used for zero-shot predictions, disentangling object-attribute pairs, image editing, retrieval and debiasing Chen et al. (2018); Vo et al. (2019); Bolukbasi et al. (2016). The embedding arithmetic has become quite popular in many VLM applications. Douze et al. (2022) shows latent text representations to exhibit geometric regularities and that text-defined transformations can be applied to image modality, demonstrating the application in image retrieval. Tewel et al. (2022) uses visual embedding arithmetic for LM-side steering to enable zero-shot image-to-text generation. A recent work Zhang et al.

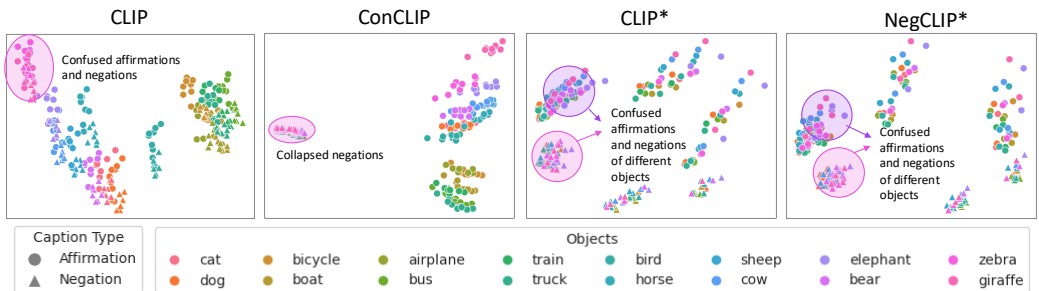

Figure 3: **Embedding space visualization with negated captions.** PCA projection of embeddings show: (1) CLIP fails to distinguish between affirmative and negative samples, (2) ConCLIP collapses all negative embeddings, (3,4) CC12M-NegFull trained models do separate out the negated embeddings from affirmative ones, however, the negated embeddings are still tightly clustered.

(2023) has used text embedding editing method to improve CLIP model's object counting ability by transferring counting knowledge between different objects. Recently, Koishigarina et al. (2025) showed that CLIP text embeddings follow compositional properties. In general, the CLIP feature space is well studied, however, it has not been explored for improving the negation understanding capabilities. Building on these insights, we propose a training-free text embedding editing method, which improves the semantic fidelity of text prompts containing negations.

## 3 PROPOSED METHOD

In this section, we present our method for embedding space correction to improve negation understanding. We first investigate some limitations of existing data centric approaches. Then we analyze, if CLIP embeddings allow arithmetic manipulations of embeddings to improve general compositional understanding? With the newfound insights, we finally introduce our approach.

### 3.1 HIDDEN LIMITATIONS OF TRAINING ON CUSTOM NEGATION DATASETS

We begin by investigating zero-shot image classification accuracies of CLIP Cherti et al. (2023) and NegCLIP Yuksekgonul et al. (2022) models fine-tuned on CC12M-NegFull. As shown in the Fig. 2(1) while these fine-tuning approaches improve the scores on NegBench-MCQ benchmarks, it comes at the cost of reduced performance on the general zero-shot classification tasks. There is a substantial drop across datasets like ImageNet1k, Caltech101, Flowers102, Stanford Cars, and Oxford Pets Parkhi et al. (2012); Krause et al. (2013); Nilsback & Zisserman (2008); Deng et al. (2009); Li et al. (2022a), suggesting that specializing for negation understanding compromises on the general representations and increases inter-class confusion as shown in Fig. 2(2).

Next, we visualize the text embeddings for different models in Fig. 3. We generate affirmative captions for several noun words such as "cat", "bicycle", "train", etc. and their negated versions such as "not cat", "not bicycle" and "not train", following the caption template from Li et al. (2025). It can be noted that the original CLIP model is not able to distinguish between the affirmative and negated examples. ConCLIP, trained on CC-Neg dataset can separate positive and negated examples but results in collapsed negated embeddings. Models fine-tuned on CC12M-NegFull Li et al. (2025) do show better discriminability between both affirmative and negated captions. However, negated embeddings from different categories appear to be tightly clustered in comparison to the positive ones, suggesting inherent limitation of these models.

### 3.2 DO CLIP EMBEDDINGS ALLOW ARITHMETIC OPERATIONS TO INTRODUCE COMPOSITIONALITY?

We consider two object nouns such as "Cat" and "Flower" and compare the embeddings for composite word "Cat and Flower" using sentence template "A photo of a {}". We denote corresponding CLIP embeddings as $e^c$, $e^f$, and $e^{cf}$, respectively. We note that the embedding $e^{cf}$, is quite similar to

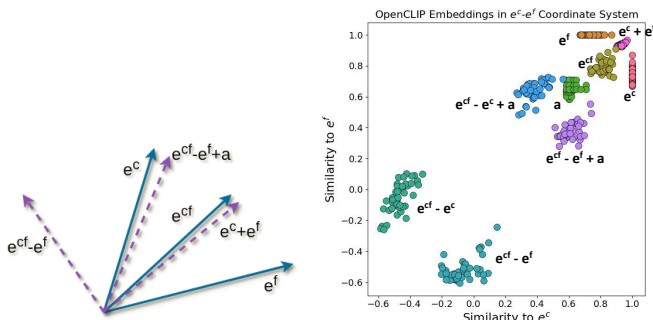

Figure 4: **Embedding space arithmetic visualization** (Left): We plot embeddings of elementary captions as $\mathbf{e^c}$ and $\mathbf{e^f}$ and plot all the other embeddings for composed captions as $\mathbf{e^{cf}}$, $\mathbf{e^c} + \mathbf{e^f}$, $\mathbf{e^{cf}} - \mathbf{e^f}$ and $\mathbf{e^{cf}} - \mathbf{e^f} + \mathbf{a}$ with respect to axis $\mathbf{e^c}$ and $\mathbf{e^f}$. Figure show that compositional embedding $\mathbf{e^{cf}}$ can be synthesized using $\mathbf{e^c} + \mathbf{e^f}$. Further the elementary embedding $\mathbf{e^c}$ can also be synthesized with the help of anchor embeddings as $\mathbf{e^{cf}} - \mathbf{e^f} + \mathbf{a}$. We note that embeddings are said to be correct if they have high cosine similarity with respect to desired embedding. For the example of cat and flower, $\mathbf{e^c} + \mathbf{e^f}$ has cosine similarity of 0.86 with $\mathbf{e^{cf}}$. Similarly, $\mathbf{e^{cf}} - \mathbf{e^f}$ and $\mathbf{e^{cf}} - \mathbf{e^f} + \mathbf{a}$ have a similarity of -0.33 and 0.83, respectively with target embedding $\mathbf{e^c}$. (Right): We also visualize embeddings for several examples.

$\mathbf{e^c} + \mathbf{e^f}$ (in terms of cosine similarity), suggesting that the semantic information in text embeddings can be added together to generate semantically accurate embeddings for the composed sentence.

We further test whether we can reconstruct the embeddings $\mathbf{e^c}$ and $\mathbf{e^f}$ using composed embedding $\mathbf{e^{cf}}$. Intuitively, simple subtraction between $\mathbf{e^{cf}}$ and $\mathbf{e^f}$ should provide candidate embedding for $\mathbf{e^c}$, however, we note that such embedding given by $\mathbf{e^{cf}} - \mathbf{e^f}$ is quite far away from the desired $\mathbf{e^c}$. We hypothesize that, in general text embeddings have some non-informative bias which offsets them into correct region in embedding space. The subtraction operation removes such common bias that results in incorrect embedding. Thus, we argue that we need to add such a bias back into the resultant embedding in the form of an *anchor* embedding, $\mathbf{a}$. One may estimate the embedding $\mathbf{a}$, as a bias computed as the mean of all embeddings on a large text corpus. One can even use embeddings of some *non-informative* words, such as "neutral", "balanced", etc. as an anchor, since these words generally convey the semantic meaning of *unbiasedness* and should be closer to all embeddings in general. Further, anchor embedding estimated in this way are dataset independent and computationally efficient. Thus we use the latter approach to estimate $\mathbf{a}$. The resultant embedding given by $\mathbf{e^{cf}} - \mathbf{e^f} + \mathbf{a}$ is indeed similar to the target embedding $\mathbf{e^c}$. We repeat above analysis for several random pairs of nouns and show all the original and recomputed embeddings in Fig. 4.

### 3.3 CAN WE IMPROVING NEGATION UNDERSTANDING WITH ARITHMETIC OPERATIONS?

Based on the analysis presented in the previous section, we now present our approach where we explicitly manipulate embeddings for better negation understanding. For a caption such as "A photo of a cat but not flower", negation understanding requires the final embedding to not have any semantic information about the negated concept "flower". However, due to affirmation bias, models may ignore the negation word and as a result the embedding is quite similar to the affirmative sentence such as "A photo of a cat and flower".

In particular, let $\mathbf{e^c}$, $\mathbf{e^{neg}}$ and $\mathbf{e^*}$ be embeddings for a given full caption $C$ with negation, the negated concept $C^{neg}$ (such as "flower" in above example) and final corrected embedding, respectively. A naive formulation for correct embedding can be given as

$$\mathbf{e^*} = \mathbf{e^c} - \mathbf{e^{neg}} + \mathbf{a} \tag{1}$$

Our experiments show that the above implementation results in an improved performance on Neg-Bench. However, we argue that such formulation is suboptimal for following reasons. First, the baseline CLIP models *may not completely ignore the negation but only struggle to eliminate it*, leading to poor negation understanding. Secondly, when fine-tuned on negation datasets, models such as NegCLIP learn to suppress negated concept more aggressively in embedding (hence improved

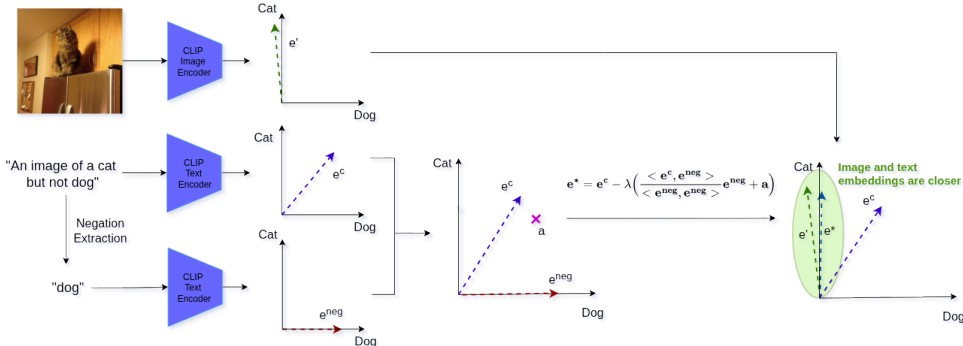

Figure 5: **Overall pipeline of our zero-shot negation correction method.** Given an input caption containing negation, we first apply rule-based splitting to separate the affirmative part ("An image of a cat but") and negated part ("not dog"). The CLIP text encoder then generates embeddings for the original caption $\mathbf{e^c}$ and the negated part $\mathbf{e^{neg}}$. Our approach then corrects the embedding for better alignment with images.

accuracy) but may not succeed in completely eliminating it. Further, the CLIP embeddings are semantically dense Bhalla et al. (2024) and represent multitude of related semantic concepts, and models may already have *removed subset of negated semantic information* in $\mathbf{e^{neg}}$ from $e^c$ correctly. For these reasons, we must first estimate the information about $\mathbf{e^{neg}}$ which is *present* in $\mathbf{e^c}$ and then only remove such semantic information. Thus, we propose the following formulation.

$$\mathbf{e}^* = \mathbf{e^c} - \lambda \Big( \frac{< \mathbf{e^c}, \mathbf{e^{neg}} >}{< \mathbf{e^{neg}}, \mathbf{e^{neg}} >} \mathbf{e^{neg}} - \mathbf{a} \Big) \qquad (2)$$

The term $\frac{< \mathbf{e^c}, \mathbf{e^{neg}} >}{\mathbf{e^{neg}}, \mathbf{e^{neg}}} \mathbf{e^{neg}}$ represents projection of $\mathbf{e^c}$ on $\mathbf{e^{neg}}$ representing common information that must be removed. This term effectively represents the *directional offset* required to generate corrected negation-aware embedding $\mathbf{e}^*$ from $\mathbf{e^c}$. We further note that, we need hyperparameter $\lambda$ to allow control over the adjustment depending on specific model and datasets.

## 3.4 OVERALL APPROACH

We summarize overall approach in Fig. 5. We define a comprehensive corpus of negation words including explicit negators (never, no, nothing, nowhere, none, not), absence indicators (absence, empty, devoid, lacks, absent), and contracted forms (n't, haven't, hasn't, can't, etc.). If none of the negator words are present in the given caption $C$, the embedding is not updated. If negator words are present instead, then we extract the negated concept $C^{neg}$ as a sequence of words affected by the negator. We then use Eq. 2 to get the updated embedding. We present the summary in Algorithm 1.

### 3.4.1 EXTRACTION OF NEGATED CONCEPT $C^{neg}$

We develop a rule-based negation scope detection algorithm, inspired from Loper & Bird (2002). The negation scope defines the sequence of words which are influenced by negation term in given caption. We begin to process the text sequentially and when a negation word is detected, the negation scope is defined depending on the type of negator and all the words inside the scope are taken as the $C^{neg}$.

---

**Algorithm 1** Embedding Correction

**Input**: Text Caption C
**Parameters**: Set $N$ of negator words
Set $A$ of anchor words
**Output**: Corrected embedding $\mathbf{e}^*$

1: **for** $w_i \in A$ **do**
2:     $a_i = $ CLIP.Encode_Text($w_i$)
3: **end for**
4: $\mathbf{a} = $ Mean($a_i$)
5: $C^{neg} = $ Null
6: **for** $n_i \in N$ **do**
7:     **if** $n_i \in C$ **then**
8:        $C^{neg} = $ *Extract_Negated_Text*($C$)
9:        break
10:     **end if**
11: **end for**
12: **if** $C^{neg} == Null$ **then**
13:     $e^* = $ CLIP.Encode_Text($C$)
14:     **return** $e^*$
15: **end if**
16: $e^c, e^{neg} = $ CLIP.Encode_Text($C, C^{neg}$)
17: $e^* = e^c - \lambda \big( \frac{e^c, e^{neg} >}{<e^{neg}, e^{neg}>} e^{neg} - \mathbf{a} \big)$
18: **return** $e^*$

---

| | MCQ | | | Retrieval | | | |
|---|---|---|---|---|---|---|---|
| | COCO | VOC2007 | MSRvtt | COCO | | MSRvtt | |
| Model | | | | R@5 | R-Neg@5 | R@5 | R-Neg@5 |
| CLIP openai | 39.3 | 38.7 | 32.1 | 54.8 | 48.6 | 50.6 | 45.8 |
| CLIP laion400M | 31.0 | 31.6 | 30.0 | 59.4 | 51.7 | 43.1 | 37.5 |
| NegCLIP | 28.7 | 30.5 | 27.3 | 68.7 | 64.4 | 53.7 | 51.0 |
| CLIP* | 54.4 | 54.8 | 44.9 | 54.2 | 51.9 | 46.9 | 43.9 |
| NegationCLIP | 36.0 | 44.1 | 34.5 | 62.2 | 61.4 | 43.0 | 41.6 |
| CLIP | 24.7 | 24.3 | 27.5 | 64.8 | 57.3 | 49.7 | 44.5 |
| Ours + CLIP | **72.5** (↑47.8) | **78.6** (↑54.3) | **50.0** (↑22.5) | - | 63.2 (↑5.9) | - | 49.0 (↑5.5) |
| NegCLIP* | 56.2 | 59.7 | 46.2 | 69.0 | 67.0 | 54.0 | 51.5 |
| Ours+NegCLIP | 69.5 (↑13.3) | 75.1 (↑15.4) | 54.1 (↑7.9) | - | **67.9** (↑0.9) | - | **52.2** (↑0.7) |

Table 1: **Performance evaluation on NegBench.** We evaluate our embedding correction method against baseline and fine-tuned CLIP models across Multiple Choice Questions (MCQ) and Retrieval tasks. (*) indicates the models fine-tuned on the CC12M-NegFull. We show improvements over original models in brackets.

We classify the negators into pre-negators and post-negators. The pre-negators are forward-scoped negators which precede the clause being negated. For example, words "empty" and "devoid" activate forward negation scope when followed by the preposition "of". So we activate the negation scope starting from the negator until a clause boundary like punctuation or conjunction is encountered. Post-negators on the other hand are backward-scoped negators, succeeding the clause being negated. For example, words "absent" and "nowhere" trigger backward negation scope when preceded by copular verbs (is/are). Therefore, when such negators are encountered, a backward negation scope is triggered until the clause boundary. We elaborate more on this in the Appendix.

## 4 EXPERIMENTS

### 4.1 BENCHMARKS, EVALUATION TASKS AND BASELINES

In this section we validate proposed approach on the benchmark datasets for negation understanding. We use NegBench Li et al. (2025) dataset comprising of 79k samples among 18 task variations. Built on top of popular datasets like COCO Lin et al. (2014), VOC2007 Everingham et al. (2010) and MSR-VTT Xu et al. (2016), this benchmark has two core tasks: Retrieval-Neg, a coarse grained retrieval task to test the model's ability to retrieve images based on a text prompt that includes both positive and negative captions; and MCQ-Neg, a fine-grained question answering, which tests the model to choose the correct description from closely related captions, including LLM-generated positive, negative and hybrid captions. We show MCQ accuracy and the retrieval performance as Recall@5 and Recall-Neg@5. Recall@5 is the performance of the model on the original captions and Recall-Neg@5 is the performance for the negated captions. We use 5% of the COCO-MCQ split as a validation data to tune $\lambda$. We also present results on CC-Neg dataset in appendix. We exclude the HardNeg-Syn Li et al. (2025) dataset since it is synthetic dataset and not publicly available.

We compare our approach directly with several pretrained (CLIP, NegCLIP) and fine-tuned models (ConCLIP, and CC12M-NegFull dataset fine-tuned variants for CLIP* and NegCLIP*). We select ViT-B/32 backbone for all models. We implement our approach on NegBench fine-tuned NegCLIP* and Laion2b pretrained CLIP model from the open_clip Cherti et al. (2023) implementation.

### 4.2 RESULTS

Table 1 shows the comparative analysis for all the models on NegBench. Our method significantly improves the performance across all MCQ datasets. For baseline CLIP model, the original accuracies on COCO, VOC2007 and MSRvtt were at chance level, 24.7%, 24.3%, 27.5%, respectively. Using proposed embedding correction, it improved to 72.5%, 78.6% and 50.0%, increasing by 47.8%, 54.3%, 22.5%. Notably, this result outperforms even the CLIP* and NegCLIP* models which are

| | MCQ | | | Retrieval | | | |
|---|---|---|---|---|---|---|---|
| | COCO | VOC2007 | MSRvtt | COCO | | MSRvtt | |
| **Model** | | | | **R@5** | **R-Neg@5** | **R@5** | **R-Neg@5** |
| SigLIP | 28.9 | 30.8 | 30.7 | 72.1 | 64.4 | 51.4 | 44.7 |
| + Ours | **61.15** (↑32.25) | **66.9** (↑36.1) | **46.8** (↑16.1) | – | **68.7** (↑4.2) | – | **47.6** (↑2.9) |
| SigLIP2 | 27.2 | 27.1 | 30.1 | 72.8 | 64.0 | 51.6 | 45.3 |
| + Ours | **66.3** (↑39.1) | **70.2** (↑43.1) | **50.1** (↑20) | – | **70.2** (↑6.2) | – | **49.4** (↑4.1) |
| AlignCLIP | 32.7 | 28.4 | 23.6 | 44.8 | 35.6 | 35.8 | 31.1 |
| + Ours | **60.1** (↑27.4) | **69.2** (↑40.8) | **44.3** (↑20.7) | – | **41.2** (↑5.6) | – | **34.7** (↑3.6) |
| TripletCLIP | 33.8 | 23.7 | 30.2 | 52.3 | 44.3 | 42.4 | 35.8 |
| + Ours | **61.8** (↑28.0) | **57.2** (↑33.5) | **46.7** (↑16.5) | – | **47.9** (↑3.6) | – | **39.9** (↑4.1) |

Table 2: Performance evaluation on other backbones show that our approach generalizes across different model architectures and training paradigm.

| Model | Accuracy |
|---|---|
| NegationCLIP | 78.3 |
| CLIP | 71.0 |
| CLIP+Ours | **79.5** (↑ 8.5) |

Table 3: Evaluation on VALSE Existence dataset, a subset of VALSE dataset.

| Model | Acc R@5 | Acc R@1 |
|---|---|---|
| CLIP | 67.4 | 34.0 |
| CLIP+Ours | **70.4** (↑ 3.0) | **41.3** (↑ 7.3) |

Table 4: Evaluation on human-annotated dataset with negation, a subset of Flickr30K dataset.

fine-tuned on CC12M-NegFull dataset with hard-negatives, highlighting that our embedding correction restores the semantic alignment of negated samples without any need of training dataset.

On the negative retrieval task (R-Neg@5) we get an improvement of 5.9% and 5.5% on COCO and MSRvtt retrievals, respectively, achieving 63.2% and 49.0% with CLIP model. When using NegCLIP*, proposed approach achieves SOTA results with 67.9% and 52.2% accuracies. Since our method only corrects the embeddings of captions with negation words, the accuracy on the positive retrieval (R@5) remains the same as baseline model.

**Generalization across different model families:** We further evaluate our approach on some of the popular VLM models such as SigLIP Zhai et al. (2023) & SigLIP2 Tschannen et al. (2025), AlignCLIP Eslami & de Melo (2024) and TripletCLIP Patel et al. (2024b) in Table 2. Our embedding correction method consistently improves the performance across all models showing strong generalization across different backbones.

**Generalization across different datasets:** We further substantiate the performance of our approach across different negation datasets other than NegBench. We first evaluate our method on a subset of VALSE Parcalabescu et al. (2022) dataset, the VALSE Existence. This dataset is similar in nature to the NegBench with 2 option MCQ task with negations. As shown in Table. 3 our method boosts CLIP model accuracy from 71.0% to 79.5% achieving best results. The evaluations so far focused on benchmarks and datasets which were synthetic and semi-automatic in nature. To further analyze the robustness of our approach, we extend the evaluation to human-annotated image caption dataset Flickr30K Van Miltenburg et al. (2016). In Table. 4 we show text retrieval accuracy for top-5 and top-1 predictions. Our approach consistently improve baseline CLIP model showing generalization to human annotations as well.

### 4.3 WHY DOES CLIP MODEL ACCURACIES IMPROVE SIGNIFICANTLY?

In this section, we visualize the effect of our method on CLIP text embeddings using PCA projections. In Fig. 6 (left), we project text embeddings of affirmative and their negated counterparts for different objects using different templates. After applying proposed corrections, the affirmative and negated samples are clearly separated along a distinct negation axis, leading to more coherent

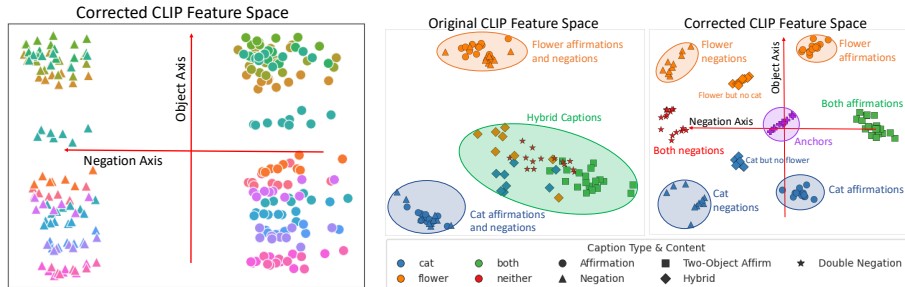

Figure 6: **Visualization of the embeddings.** (Left): For affirmative and negated captions, corrected CLIP embeddings show separation along negation axis and maintains original semantic grouping. (Center): Original CLIP space with complex captions: single object affirmations/negation, double affirmations/negations and hybrid (Right): Proposed embeddings are well separated and organized.

| Caption Type | Confidence Score | | |
|---|---|---|---|
| | CLIP | NegCLIP* | Ours |
| Affirmative | 0.45 | 0.49 | 0.65 |
| Negated | 0.16 | 0.31 | 0.64 |

Table 5: Average confidence score between image and caption (both affirmative and negated).

| Model | Positive ↑ | Negative ↓ |
|---|---|---|
| ConCLIP | 83.8 | 57.2 |
| NegCLIP* | 86.8 | 78.7 |
| CLIP | 89.5 | 89.8 |
| Ours+CLIP | **89.5** | **10.2** |

Table 6: Zero-shot accuracy on CIFAR10 with positive and negative (distractor) captions.

and aligned clusters. These embeddings are also well organized by class groups, indicating that our method is able to not only disambiguate negation, but also preserve the semantic class structure similar to affirmative embeddings. In comparison, CLIP* and NegCLIP* embeddings (Fig. 3) for negated captions often lead to tight clusters which lead to poor accuracies.

We further extend this analysis to more complex caption types. We show hybrid prompts ("a cat and not a flower"), two object affirmations ("a cat and a flower") and double negations ("neither cat nor flower"), along with the single object captions. In the original CLIP space (center), the affirmations and negations have very high overlap while all the complex sentences collapse to the same cluster at the center. However, after applying the corrections, all the different categories are separated into semantically interpretable clusters, separating different negations and semantic classes, both. For example, the double affirmations lie between single object affirmations while the double negations lie between single object negations and form a cluster along the same virtual negation axis. Further, the hybrid captions lie between affirmative and negated clusters for both classes.

These plots highlight the compositional consistency and semantically meaningful structure introduced by our method, explaining why proposed approach performs so well. It not only corrects the simple negation but also generalizes well to hybrid negations involving multiple objects.

### 4.4  HOW WELL DO CLIP MODELS ALIGN THE NEGATED CAPTIONS WITH IMAGES?

We analyze the cross-modal alignment between text and images using probability scores between ground truth image-text pairs on COCO NegBench-MCQ. We show the average score for affirmative and negated ground truth captions separately in Table. 5. We note that CLIP model achieves the score of 0.45 for positive captions but struggle to align negated captions with corresponding images and achieves the score of 0.16. The fine-tuned model NegCLIP* achieves the score of 0.31 for negated captions which explains the improved accuracies. When our proposed embedding correction is applied, the CLIP score improves from 0.16 to 0.64 showing improved alignment between negated text and corresponding images. Further, the score for affirmative captions improves from 0.45 to 0.65, due to better understanding of incorrect negated captions, thus reducing model confusion.

In Table. 6, we also validate image classification score on CIFAR10 dataset by using negated class template "This is not a photo of {class}." as a distractor to see how well the models push away

distractor text embedding from images. Surprisingly, CLIP with corrected embeddings achieves 10.16% accuracy which is very close to random chance.

These analyses reveal that proposed negation-aware embeddings are well aligned with images in the presence of negation, when compared to fine-tuned models, explaining the superior performance.

### 4.5 Ablation: Comparing different embedding correction methods

We compare the naive formulation in Eq. 1 with formulation in Eq. 2 for different values of $\lambda$ on COCO-MCQ split of Neg-Bench in Fig. 7. We also show results when excluding the anchor term $\mathbf{a}$. We note that, for Neg-CLIP*, the optimal value for $\lambda$ is 0.3 whereas for CLIP is it 1.9. This result is expected since Neg-CLIP* already fine-tuned and separates negated captions into separate cluster, thus requires only minor adjustment whereas original CLIP needs large offset to correct the embeddings.

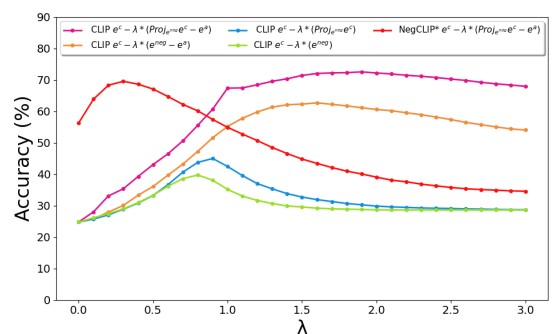

Figure 7: We show accuracies for different values of $\lambda$. For CLIP we show results for different approaches. Projection of $\mathbf{e^c}$ on $\mathbf{e^{neg}}$ is denoted by $\mathbf{Proj_{e^{neg}}}$.

## 5 Limitations and Future Work

While our training-free, embedding correction method achieves state-of-the-art results and successfully mitigates the catastrophic forgetting associated with fine-tuning approaches, the overall system retains two primary limitations that define clear paths for future research. The most significant is the *syntactic brittleness of the negation extraction approach*. The reliance on a rule-based component for identifying the precise scope of the negated concept ($\mathbf{C^{neg}}$) is inherently limited. Specifically, this approach struggles with complex syntactic structures (e.g., long-distance dependencies like "A large, decorative, but clearly *not* present, table."), implicit negators (e.g., quantifiers like "few people"), and ambiguous scope arising from multiple negations or complex conjunctions. We emphasize that this is a bottleneck of negation extraction step only and not a flaw in our core vector arithmetic approach. A clear path for generalization is the integration of a *lightweight, specialized, sequence tagging NLP model* (e.g., fine-tuned DistilBERT) to ensure robust scope detection while maintaining the core zero-shot VLM approach. A second limitation concerns the *simplicity of current benchmarks* (NegBench, VALSE Existence, CC-Neg). These datasets primarily evaluate *explicit, simple negation* focused on single objects. The remarkable success of our training-free method suggests these benchmarks chiefly test the disentanglement of explicitly marked concepts. To truly test VLM logical reasoning, future benchmarks must incorporate complex logical structures (e.g., double negation) and implicit negation.

## 6 Conclusion

We analyze properties of CLIP models on the task of negation understanding and note that despite available training datasets, these models struggle to fully understand negations. We show that CLIP text embeddings can be manipulated to add or remove semantic information. Using this idea, we present a rule-based approach to explicitly remove semantic information about negated concept from text embeddings. Our approach improves baseline model and outperforms even the best models by a wide margin. These results highlight the limitations of current approaches which are largely data driven, and suggest that future works should focus on enforcing compositional properties explicitly into the framework. For example, along with contrastive loss, one can additionally use our approach to regularize learned text embeddings for negated captions which may help boost negation understanding. We hope to motivate research community to pursue efforts in this direction.

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

# A APPENDIX

## A.1 EXTRACTING NEGATED CONCEPTS $C^{neg}$

Extracting the negated concepts from given caption is a very crucial step, and therefore, we develop a rule-based negation scope detection algorithm that identifies negation cues and selects all words within the scope as $C^{neg}$. Some negators require specific context, for example the words "absent" and "nowhere" trigger backward negation scope when preceded by copular verbs (is/are), whereas "empty" and "devoid" activate forward negation scope when followed by the preposition "of". Therefore, we classify the negators into pre-negators and post-negators. We define a comprehensive list of negation words in each category as shown in Table. 7 and 8. For a given caption, we process the texts sequentially and check the presence of a negation word from both categories and then select part of the caption as $C^{neg}$ accordingly.

The pre-negators are forward-scoped negators which means that they precede the clause they are negating, so we activate the negation scope starting from the negator until a *clause boundary* like punctuation or conjunction is encountered. We explicitly define the clause boundary as punctuation marks, coordinating conjunctions, contrasting conjunctions, and subordinating conjunctions. Punctuation marks (periods (.), commas (,), semicolons (;)), and contrasting conjunctions (but, although, though, however, nevertheless, yet, even so, while, whereas) typically terminate negation scope, whereas coordinating conjunctions (and, so, for) and subordinating conjunctions (because, since, as, if, unless, until) generally don't terminate the negation scope. Post-negators on the other hand are backward-scoped negators, succeeding the clause they are negating. Therefore, when such negators are encountered, a backward negation scope is triggered until the clause boundary. In this way, we are able to extract the negated concept from given caption for both pre and post negations.

We present the template for extracting $C^{neg}$ below

**Caption Template for processing pre-negators**
<caption text> <pre-negator> <words within the negation scope> <clause boundary> <remaining Caption>

**Caption Template for processing post-negators**
<Caption text> <clause boundary> <words within the negation scope> <post-negator> <remaining Caption>

| Pre-Negators |
| --- |
| does not, lack, no, not, never, nothing,noone, none, nowhere, without misses, neither, nor, absence/devoid/empty of, hasn't, haven't, doesn't, don't |

Table 7: The list of pre-negators terms used.

| Post-Negators |
| --- |
| is/are/was/were nowhere/not/absent/never/lacking/missing/neither/nor, absence/devoid/empty of, |

Table 8: The list of post-negators terms used.

| Caption $C$ | Negator | Negated Part $C^{neg}$ |
|---|---|---|
| **Pre-Negation** | | |
| This image does not include any cups | does not | include any cups |
| This image features a chair but lacks a handbag. | lacks | a handbag |
| This image features a person, with no dining table in sight. | no | dining table in sight |
| The image depicts a person, without a backpack. | without | a backpack |
| Note the absence of a traffic light in this image. | absence of | a traffic light in this image |
| This image features a chair, but is empty of people. | empty of | people |
| No pizza is included in this image. | no | pizza is included in this image |
| This image shows a bowl but doesn't include a refrigerator. | doesn't | include a refrigerator |
| | | |
| **Post-Negation** | | |
| Skis are not visible in this image. | are not | skis |
| A sink is nowhere to be seen in this image | is nowhere | a sink |
| A cat is present in this image, whereas a person is not. | is not | a person |
| A cup is absent from this image. | is absent | a cup |
| A TV is present in this image, while chairs are not | are not | chairs |

Table 9: Sample captions and corresponding negators with negated concept $C^{neg}$ are shown.

## A.2 COMPARISON WITH OTHER RULE BASED PARSER:

The NLP community has proposed several deterministic negation detection methods (e.g., NegEx, DEEPEN). Our current rule-based approach is significantly motivated by established clinical NLP tools like NegEx. The evaluation on NegBench-COCO split for CLIP backbone using our custom VLM-focused rules and an off-the-shelf NegEx implementation are shown in Table. 10. As can be noted, NegEx based caption parsing resulted in lower performance of 61.3% compared to our method 72.5%. We note that we use NegEx simply as an off-the-shelf component here, to parse the negated concept from the caption. The NegEx "termsets" does not include some of the common negations in natural language and as a result fails to parse many captions correctly leading to lower performance. For example, the pre-negation words such as 'lacks', 'excludes', 'won't', 'devoid of', 'absent', etc. are absent in NegEx. Similarly, post-negator words such as 'is not', 'is nowhere' and 'is absent' are missing. We believe that, incorporating these negator words in the NegEx termsets will lead to better performance.

| Method | Acc. |
|---|---|
| Ours | 72.5 |
| NegEx | 61.3 |

Table 10: We compare our rule based negation parser with off the shelf NegEx library.

# B APPENDIX

## B.1 RESULTS ON CC-NEG

We compare our approach with other methods on the CC-Neg dataset as shown in Table 11. Our method is easily able to outperform the existing methods, attaining the highest accuracy of 99.79%, even outperforming the Con-CLIP model, fine-tuned on the CC-Neg dataset.

## B.2 EVALUATION ON DIFFERENT MODEL PRETRAINING AND MODEL SIZES

To assess the effect of model size on our model's effectiveness, we evaluate a range of CLIP models pretrained on the laion dataset in Fig. 8. We see that our method yields significant improvement across all model sizes.

| Model | CCNeg Acc |
|---|---|
| CLIP openai | 65.70 |
| NegCLIP | 62.63 |
| FLAVA | 58.93 |
| BLIP | 62.31 |
| Con-CLIP | 99.70 |
| **Con-CLIP+Ours** | **99.79** |

Table 11: MCQ accuracy (%) comparison on the CC-Neg benchmark. Our zero-shot method outperforms all baseline models including the fine-tuned Con-CLIP model, specifically trained on CC-Neg data. This shows the effectiveness of our approach when compared to the models with domain-specific fine-tuning.

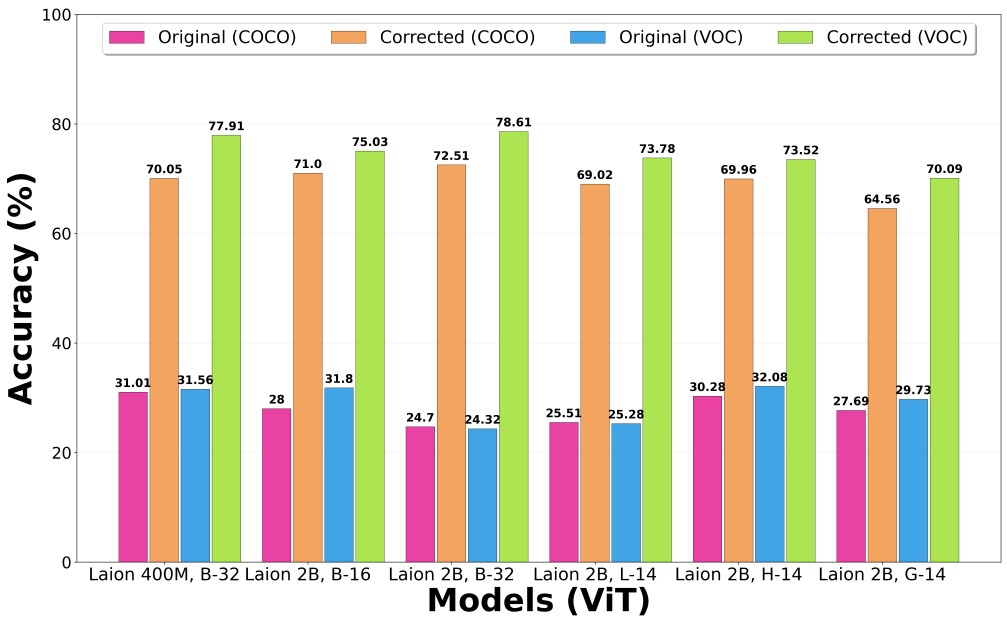

Figure 8: Effect of model architectures (B-16, B-32, L-14, H-14, G-14) and pre-training data size (LAION-400M vs LAION-2B).

Further, to understand the role of pretraining data size, we also compare the performance of ViT-B/32 model with two pretrained weights, laion-400M and laion-2B. While the original performance of the model trained on the larger laion-2B dataset is worse (most likely due to the overfitting on affirmations and lack of exposure to negated samples), our method improves both models consistently and laion-2B model achieves best result indicating learning of richer semantic information.

### B.3 EFFECT OF DIFFERENT ANCHOR CHOICES

We compare various choices of our anchor embedding **a** on the NegBench COCO MCQ in Table 12. We use a few candidate words which are commonly used in natural language to show unbias and impartial opinions. We then compare results when using average of all these words embeddings as an anchor. The most ideal choice, mean of all embeddings from text encoder, is also considered. We compute the mean of all captions in COCO MCQ NegBench dataset. We note that, this mean is used as an oracle anchor to compare performance of other words as a choice for anchor. We note that as long as we choose semantically netural word as the anchor there is not much variation in the evaluation results.

Next, we also compare results when using random nouns or other words with *semantic inclination* such as "extreme", "positive", etc. We note that, results drastically drop with such a choice of biased anchor.

| Anchor Word | Acc |
|---|---|
| 'neutral' | 72.51 |
| 'balanced' | 72.54 |
| 'unbiased' | 72.42 |
| 'fair' | 72.44 |
| Mean of Above | 72.52 |
| Mean of NegBench-MCQ | 72.53 |

Table 12: NegBench MCQ accuracy on COCO split. We show results when using different anchor embeddings. When anchor words with semantic meaning of *neutralness* are used, we achieve good accuracies.

| Anchor Word | Acc |
|---|---|
| 'cat' | 50.78 |
| 'dog' | 50.69 |
| 'biased' | 54.67 |
| 'positive' | 53.93 |
| 'extreme' | 54.31 |
| 'right' | 52.16 |

Table 13: NegBench MCQ accuracy on COCO split. We show results when using different anchor embeddings. When using arbitrary noun words or words with specific inclination such as the above words, accuracy dramatically drops.

### B.4    RECURSIVELY ADDING ANCHOR EMBEDDINGS

We analyze the effect of anchor embedding $\mathbf{a}$ on corrected embeddings $\mathbf{e^{cf}} - \mathbf{e^{f}}$ and $\mathbf{e^{cf}} - \mathbf{e^{c}}$. The first half shows that without anchor embedding the corrected embedding $\mathbf{e^{cf}} - \mathbf{e^{f}}$ is dissimilar to desired $\mathbf{e^{c}}$ and after using anchor $\mathbf{e^{cf}} - \mathbf{e^{f}} + \mathbf{a}$, its similarity improves. However, further adding anchor terms as in $\mathbf{e^{cf}} - \mathbf{e^{f}} + \mathbf{2a}$ and $\mathbf{e^{cf}} - \mathbf{e^{f}} + \mathbf{3a}$ makes it more similar to the other embeddings $\mathbf{e^{f}}$ i.e. it add more and more confusion into the embeddings. A similar trend is observed for $\mathbf{e^{f}}$ when repeated anchor embeddings are added into $\mathbf{e^{cf}} - \mathbf{e^{c}}$. This result is expected since anchor embedding is supposed to be unbiased and hence more closer to all the other embeddings. Thus, repeatedly adding anchor embeddings will remove semantic information from original embedding and will bring it closer to other embeddings.

| Target | | $\mathbf{e^{c}}$ | $\mathbf{e^{f}}$ | $\mathbf{e^{cf}}$ |
|---|---|---|---|---|
| $\mathbf{e^{c}}$ | $\mathbf{e^{cf}} - \mathbf{e^{f}}$ | -0.07 | -0.52 | 0.10 |
| | $\mathbf{e^{cf}} - \mathbf{e^{f}} + \mathbf{a}$ | 0.61 | 0.38 | 0.57 |
| | $\mathbf{e^{cf}} - \mathbf{e^{f}} + \mathbf{2a}$ | 0.64 | 0.54 | 0.54 |
| | $\mathbf{e^{cf}} - \mathbf{e^{f}} + \mathbf{3a}$ | 0.64 | 0.59 | 0.52 |
| $\mathbf{e^{f}}$ | $\mathbf{e^{cf}} - \mathbf{e^{c}}$ | -0.47 | -0.05 | 0.13 |
| | $\mathbf{e^{cf}} - \mathbf{e^{c}} + \mathbf{a}$ | 0.38 | 0.64 | 0.56 |
| | $\mathbf{e^{cf}} - \mathbf{e^{c}} + \mathbf{2a}$ | 0.52 | 0.67 | 0.53 |
| | $\mathbf{e^{cf}} - \mathbf{e^{c}} + \mathbf{3a}$ | 0.56 | 0.67 | 0.52 |

Table 14: We analyze effect of anchor embedding on the quality of corrected embedding.

| Image | Ground Truth Caption | Caption selected by Neg-CLIP* | Caption selected by our alg. |
|---|---|---|---|
|  | Note the absence of a chair in this image. | No wine glass is included in this image. ✗ | Note the absence of a chair in this image. ✓ |
|  | A cup is nowhere to be found in this image. | A cup is present in this image, but there is no pizza. ✗ | A cup is nowhere to be found in this image. ✓ |
|  | This image contains a bowl but lacks a cup. | A bowl is not present in this image. ✗ | This image contains a bowl but lacks a cup. ✓ |
|  | A chair is absent from this image. | No fork is present in this image. ✗ | A chair is absent from this image. ✓ |
|  | No person is present in this image. | A carrot is not included in this image. ✗ | No person is present in this image. ✓ |
|  | A dining table is not present in this image. | There is no hot dog in this image. ✗ | A dining table is not present in this image. ✓ |
|  | The image showcases both a refrigerator and a cup. | This image does not contain a refrigerator. ✗ | The image showcases both a refrigerator and a cup. ✓ |
|  | A car is absent from this image. | No bowl is present in this image. ✗ | A car is absent from this image. ✓ |
|  | This image contains a book, with a notable absence of a chair. | A chair is included in this image, but there is no book. ✗ | This image contains a book, with a notable absence of a chair.✓ |
|  | This image contains an orange, but a person is nowhere to be seen. | No orange is included in the image. ✗ | This image contains an orange, but a person is nowhere to be seen. ✓ |

Table 15: Qualitative comparison of caption outputs for NegCLIP* model and our algorithm.

## B.5 QUALITATIVE ANALYSIS

In Fig. 15 we show some qualitative examples where NegCLIP* model fails to understand negation in the input captions. For example, in row 1, NegCLIP* model possibly aligned 'wine' text features with image contents and ignored the 'no' completely. Our approach corrected this embedding which resulted in this caption *moving away* from corresponding image embedding. Similarly, for second row, the word 'no pizza' resulted in incorrect alignment with image embedding but our method explicitly corrected the text embedding.

