# OpenReview forum: "Seeing What’s Not There: Negation Understanding Needs More Than Training"
_ICLR.cc/2026/Conference — ICLR 2026 Poster_

### Official Review · Reviewer_aYJg · 2025-10-21

**Soundness:** 3
**Presentation:** 4
**Contribution:** 3
**Rating:** 8
**Confidence:** 4

**Summary:**

- This paper investigates the persistent difficulty of VLMs in handling linguistic negation. Rather than relying on large-scale negation datasets and fine-tuning, the authors propose a training-free embedding correction method that manipulates CLIP’s embeddings directly. By identifying the negated concept via rule-based parsing and removing its semantic component through a projection-based subtraction (plus a neutral anchor addition), the approach aims to improve negation understanding.
- Experiments on NegBench and CC-Neg show large performance gains over CLIP and fine-tuned NegCLIP baselines. The authors also provide some analyses to show intuitions onto why their proposed method works.

**Strengths:**

- The idea of improving negation understanding through direct, post-hoc manipulation of text embeddings is novel and unconventional to the more standard data-centric strategies. The approach offers a new angle for thinking about compositional semantics in VLMs.
- The reported improvements on NegBench are substantial, even surpassing models fine-tuned on specialized negation datasets.
- The embedding-space visualizations (PCA plots) and ablation studies provide qualitative and quantitative insights into how the correction affects embedding geometry.
- The approach could inspire further work on embedding-space editing, compositional regularization, and interpretability.

**Weaknesses:**

- The main concern is the lack of theoretical grounding. The main motivation that negation corresponds to a linear operation in embedding space is not theoretically or empirically justified. CLIP embeddings are known to be highly entangled, and the projection subtraction used here lacks formal motivation beyond analogy to word-vector arithmetic. The choice of "neutral" words also seem arbitrary. This is totally fine as the emperical results are good, but the authors should be careful with their claims.
- The method’s success on simple caption pairs may not translate to more nuanced natural language understanding (more complex negation patterns).

**Questions:**

- Strange citation bug, e.g. line 143, Similarly TripletCLIP...
- Consider using an off-the-shelf tool for negation detection. In fact, your algorithm is quite similar to NegEx (https://pypi.org/project/negspacy/).
- Improvements of 40–50% absolute accuracy for a post-hoc method seem implausibly high. Did the authors reimplement the CLIP baselines and confirm that those results are comparable with previous works of using CLIP (and NegCLIP) on those benchmarks?

---

> ### Author Response · Authors · 2025-11-22
> **Authors’ Response to Reviewer aYJg**
>
> We appreciate the reviewer's comprehensive summary, positive assessment, and insightful critique regarding the theoretical foundation of our work. We are encouraged by the recognition of our method's novelty and the substantial empirical gains.
> Here is our response to the weaknesses and questions:
>
> **W1: Lack of Theoretical Grounding for Linear Operation**
> We agree that the claim that "negation corresponds to a linear operation in embedding space" requires careful framing. We will moderate our claims and provide a clearer, empirically-driven justification in the revised paper. We acknowledge that our projection-subtraction and anchor-addition strategy is motivated by empirical results in our experiments. We like to add that the arithmetic manipulation of CLIP embeddings is exploited in the literature. Notably in Zerocap [3] (as noted by reviewer G9hW), authors add or subtract image embeddings and predict the text for residual semantic concept. Similarly, in [4] image embeddings are edited to improve object counting abilities and in [5] text embeddings are manipulated to reveal sematic direction to for edited image. These works, along with our own experimental findings, present strong empirical evidence that clip embeddings do enable linear operations for semantic manipulation of embeddings. We have briefly addressed this topic in Section 2.3. We will add more discussion in the paper and motivate our hypothesis from empirical and experimental perspective.
>
> **W2: Success on Simple Pairs vs. Complex Negation**
> We acknowledge the reviewer's concern that current benchmarks present simple negations and to truly validate the effectiveness of our embedding correction approach we need to evaluate it on complex sentences with negations. However, there are no such datasets present in the literature and thus we can not validate our method on such complex negation patterns. The complex language syntax presents two challenges: (1) Negation parsing and (2) Negation aware embedding correction.
> **1. Negation Parsing:** The modular nature of proposed approach allows drop in replacement of new negation parsing module such as NegBERT [2].
> **2. Negation aware embedding correction:** We have shown strong performance of embedding correction approach on different backbones (CLIP, NegCLIP, SigLIP ½, ALIGN, TripletCLIP, etc.) (refer to response to G9hW), on different semi-automatic benchmarks (NegBench MCQ and retrieval, CC-Neg, VALSE) and on human annotated negation dataset such as Flickr30K subset (refer to response to u7Qg for results). These results suggest that proposed method is quite robust showing strong performance and achieving state-of-the-art results. We hope this clears reviewer's concerns.
>
> **Q1: Strange Citation Bug**
> Thank you for careful attention. However, we do not exactly understand what is the exact bug. Kindly let us know more about the issue so that we can correct the formatting in the final version of the paper.
>
> **Q2: Use of Off-the-Shelf Tool (NegEx/negspacy)**
> This a thoughtful suggestion. As detailed in our rebuttal to Reviewer u7Qg, we have already conducted a comparative evaluation on NegBench-COCO between our rule based and an off-the-shelf implementation of NegEx. We found that NegEx termset misses some of the negation keywords and hence gives poor results. After updating the termset list, NegEx may give results similar to ours. Please refer to our detailed response to Reviewer u7Qg.
>
> **Q3: Concerns over High Absolute Accuracy Gains and Baselines reproducibility:**
> We understand the skepticism regarding a post-hoc method achieving such dramatic improvements. The significant gains are a consequence of the inherent failure mode of CLIP due to Affirmation bias. Standard CLIP exhibits a strong "affirmation bias" and often ignores negation and thus performs at a chance level. We note that similar chance level performances are observed by NegBench team [1]. Their NegCLIP model which is pre-trained on ARO dataset achieves 28.6% and 27.3% accuracies (slightly above chance level) on NegBench dataset with COCO and MSRvtt splits. After retraining, their model improves to 56.2% and 46.2% achieving +27.6% and +18.9% absolute boost. And when using our post-hoc embedding corrections, results further go up to 69.5% and 54.1%. The results obtained on our baseline CLIP models follow the same trend with original results of 24.7% and 27.5% and post-hoc results of 72.5% and 50.0%. We do note that original results of CLIP are different from the one reported in [1], possibly due to different pretrained weights. We used the weights with tag 'laion2b_s34b_b79k' in all our experiments.

---

> > ### Author Response · Authors · 2025-11-22
> > **Authors’ Response to Reviewer aYJg**
> >
> > References:
> > 1. Li, G., et al. (2025). "Vision-language models do not understand negation." Proceedings of the IEEE/CVF Conference on Computer Vision and Pattern Recognition (CVPR).
> > 2.  Khandelwal, A., et al. (2020). "NegBERT: A Transfer Learning Approach for Negation Detection and Scope Resolution." Proceedings of the twelfth language resources and evaluation conference.
> > 3. Tewel, Yoad, et al. (2022) "Zerocap: Zero-shot image-to-text generation for visual-semantic arithmetic." Proceedings of the IEEE/CVF conference on computer vision and pattern recognition.
> > 4. Zang, R., et al. (2023) "Zero-shot improvement of object counting with clip." NeurIPS Workshop.
> > 5. Couairon, G., et al. (2022) "Embedding Arithmetic of Multimodal Queries for Image Retrieval." Proceedings of the IEEE/CVF Conference on Computer Vision and Pattern Recognition (CVPRW).

---

> ### Comment · Reviewer_aYJg · 2025-11-25
> **Response to authors**
>
> Thank you for your efforts in addressing the concerns. As my concerns are more about presentation and clarification, I believe the initial positive rating is fair and will keep my scores.

---

> > ### Author Response · Authors · 2025-11-25
> > **Authors’ Response to Reviewer aYJg**
> >
> > Thank you for your comment.

---

### Official Review · Reviewer_dika · 2025-10-30

**Soundness:** 2
**Presentation:** 2
**Contribution:** 2
**Rating:** 6
**Confidence:** 3

**Summary:**

The paper points out that visual-language models such as CLIP have weak understanding of negation (no/not/without), often exhibiting affirmative bias. The authors propose a zero-training text embedding correction: rules extract the negated concept, estimate and subtract the "negation direction" from the embedding, significantly improving negation understanding without fine-tuning the model, and avoiding the decline in generality caused by fine-tuning the negation data.

**Strengths:**

The method requires no hard negation data collection or model fine-tuning; it only needs to correct the "negation direction" in the CLIP text embedding, resulting in extremely low deployment costs.

On NegBench, it improved the MCQ of CLIP from approximately 25.5% to approximately 67.0%, and the retrieval rate from approximately 50.9% to approximately 56.1%; it also brought considerable gains to the fine-tuned NegCLIP.

**Weaknesses:**

The method assumes that "negation" is approximately linear and directional in the embedding space, but the semantics of "missing" different concepts/attributes are not necessarily collinear; using the global d_global may be ineffective or even lead to incorrect offsets for fine-grained concepts.


Have you tried non-CLIP text towers (such as BLIP, SigLIP)? Even small-scale experiments should demonstrate the generality of the architecture.

How does the average overhead compare to standard CLIP text encoding? Is it easy to batch process to support large-scale retrieval? Please include a small performance table.

If λ is selected on COCO-MCQ, how would the retrieval be affected by fixing that λ to VOC/MSRVTT MCQ?

Rule-based negation scope is fragile. Negation scope extraction (Algorithm 1, page 6; Tables 6–8 in Appendix A.1, pages 12–14) has limited coverage of nested clauses, multiple negations, implicit negations (such as “few/barely”), quantifiers/comparison structures, etc. The main experiments are primarily based on NegBench patterns; generalization to real open-domain instructions remains uncertain.

**Questions:**

see the weaknesses

---

> ### Author Response · Authors · 2025-11-22
> **Authors’ Response to Reviewer dika**
>
> We sincerely thank the reviewer for their thorough review, positive assessment, and valuable constructive feedback regarding low deployment costs, superior performance. We address your additional concerns below:
>
> **W1. Concerns over  linearity of negation direction and over d_global being ineffective for fine-grained concepts.**
> **1. Core Approach:** We briefly reiterate our approach here. We first extract the negated sub text $C_{neg}$ using rule based algorithm and then correct the embedding of original text using Eq. 2. This operation will correct the embeddings $C$ using directional offset for each input text. This offset direction is opposite to the embedding of $C_{neg}$ as this is the exact semantic information we need to eliminate from original embeddings. However, we agree that embedding $C_{neg}$ may represent many fine-grained semantic concepts and their individual presence/strength in the original embedding $C$ may vary. This warrants a fine grained magnitude control over each semantic direction. Thus we estimate the projection of $C$ on $C_{neg}$ as offset (Eq.2), allowing fine-grained control over each concept. Our empirical findings show effectiveness of this strategy over simply using $C_{neg}$ (Eq.1).
> **2. Motivation:** The hypothesis that CLIP embeddings (for texts and images) can be manipulated directly to adjust semantic information is not new. We discuss some of the works in this direction in Section 2.3. For example, [1] presents SIMAT benchmark to test image retrieval by editing image and text embeddings. We hope this clarifies the concern reviewer have.
>
> [1] Embedding Arithmetic of Multimodal Queries for Image Retrieval, CVPRW 2022
>
> **W2. experiments with non-CLIP text towers**
> We have tested our approach on SigLIP 1 & 2, ALIGN and TripletCLIP models (refer to the response G9hW). Our method generalizes right out of the box on these architectures and boosts their performance significantly w/o any additional parameter tuning.
>
> **W3. Overhead compare to standard CLIP text encoding**
> Computational overhead is an important parameter to judge the effectiveness of any approach. We apologize for overlooking this important aspect from our analysis. Our approach analyses each input text for the presence of negation word (small look up table based search) and then parses negation scope (light weight template matching process). If no negation is detected, we generate text encoding for the caption. If negation is detected then additional embedding is computed for sub-caption withing the negation scope. In summary, we need two CLIP Text Encoder forward passes when input caption has negation. Thus worst case complexity is twice (assuming all captions have negations) that of original CLIP approach. We will elaborate on this point in the appendix.
>
> **W4. Effect of tuning λ**
> We observe in our experiments that optimal λ is model specific and not dataset specific. When tuning λ for NegCLIP model on full MSRvtt-MCQ, the optimal value changed only slightly (0.3 -> 0.25) with negligible changes to the accuracy (54.1 -> 54.8). Similar trend is observed for CLIP model for other datasets. The retrieval tasks are quite stable with respect to different λ values and give same accuracy for broad range of λ values including one selected from COCO-MCQ.
>
> **W5. Concerns over rule based negation parser and generalization over other datasets:**
> We agree with the reviewer on these limitations. However, we note that, rule based negation detection is not the only contribution of the present work. The primary contribution is the embedding correction approach which enables zero-shot negation aware text embedding giving robust performance boost across different backbones, datasets, etc. Further, many contemporary works rely on simple benchmarks with explicit negations (NegBench, VALSE, CC-Neg), same as ours. We do acknowledge that research community needs new benchmark datasets with implicit negations and complex patterns to truly evaluation negation awareness of VLMs. Our work shares these limitations with existing methods. However, the modular design of our approach allows future extensions by upgrading rule based negation extraction. Notwithstanding the above, our rule based negation detection is quite effective and performs better than existing approach (NegEx) and generalizes to human annotated datasets (Flickr_30K) as well (Please refer to the author's repsonse submitted to reviewer u7Qg).

---

> > ### Comment · Reviewer_dika · 2025-11-28
> >
> > The author's reply resolved my question, so I chose to increase my score.

---

> > > ### Author Response · Authors · 2025-12-01
> > > **Authors’ Response to Reviewer dika**
> > >
> > > Thank you for raising the score. We really appreciate it.

---

### Official Review · Reviewer_YhKj · 2025-11-01

**Soundness:** 4
**Presentation:** 3
**Contribution:** 3
**Rating:** 4
**Confidence:** 4

**Summary:**

This paper addresses VLMs like CLIP’s poor negation understanding (e.g., ignoring "no"/"without"). Unlike data-centric methods using hard-negative datasets, it proposes a zero-shot, training-free embedding correction approach. In particular, the authors propose a rule-based method to extract negated concepts, then a correction formula removes their semantic info from embeddings (using directional offset and anchor embedding).

Evaluated on NegBench, it boosts CLIP’s MCQ accuracy from 25.5% to 67.0% and retrieval accuracy from 50.9% to 56.1%, outperforming fine-tuned models like NegCLIP.

**Strengths:**

S1. This paper is well written and is easy-to-follow.

S2. The performance improvement is clear.

S3. Sufficient visualizations are provided.

**Weaknesses:**

W1. The performance evaluation side is insufficient. Limited methods are compared. VLMs are mentioned but no further study is conducted.

W2. Typos should be corrected.

W3. According to Alg. 1, a set of negator words is utilized. The correctness of these words should be discussed. Besides, the quality of the generated embedding should be studied.

See questions for more details.

**Questions:**

Q1. Only CLIP is discussed. How about applying to SigLIP and SigLIP2? These encoders are more welcomed in practical use.

Q2. MLLMs use CLIPs to encode features. MLLM benchmarks can also be used to evaluate the performance of VLMs. How about using your method to train a LLaVA-1.6 model. Will the general VQA performance boosted?

Q3. In Alg.1, return missing ' '; ’’ should be `` ''

Q4. Will synonyms influence your algorithm? What is the quality and diversity of corrected embedding?

Q5. What is the general retrieval, VQA, classification of the proposed method?

**Details Of Ethics Concerns:**

Overall, I think this is a useful paper.

I will update my rating based on the rebuttal and other reviewers' comments.

---

> ### Author Response · Authors · 2025-11-22
> **Authors’ Response to Reviewer YhKj**
>
> Dear Reviewer,
> We sincerely thank you for your assessment and feedback. We are happy that you acknowledged the performance improvements shown by our method. We address your concerns and questions below:
>
> **W1 Q1. Concerns over performance evaluation on additional backbones:**
> We have evaluated our method on few other backbones such as **SigLIP 1 & 2, ALIGN and TripletCLIP (refer to the response submitted to reviewer G9hW)**. Our method consistently boost performance of all the methods significantly for both MCQ and retrieval tasks right out of the box. We have further evaluated our approach on additional datasets such as **VALSE Existence** and publicly available human annotated negation dataset **Flickr_30K (refer to the response submitted to reviewer u7Qg)**. These additional evaluations further substantiate the robustness of our method and we hope that this will resolve the concerns of the reviewer.
>
> **W2 Q3. Typos**
> We apologize for editing errors. We will thoroughly check and revise the manuscript before camera ready version if paper is accepted.
>
> **W3. Correctness of negator words and the quality of embeddings**
> 1. **Concerns over the list Negator words:** We have tried to include exhaustive list of negator words and patterns commonly found in English literature. For this task we prompted the LLM AI assistant such as Gemini or ChatGPT to generate list of negator words. We further manually screen the list and re-prompt the LLM models for any missing words. Further, as suggested by other fellow reviewers (u7Qg and aYJg) there are other off the shelf tools such as NegEx and DEEPEN available for the same task. We compare our negation parsing algorithm with NegEx (refer to the response submitted to u7Qg) and found that our list of negators is actually quite exhaustive. The NegEx, designed for negation detection for clinical data analysis actually misses many common negators which we have included. This experiment validates the correctness of negator words used.
> 2. **Concerns over quality of Embedding:** We agree with the reviewers' concern over the quality of the generated embedding. We believe the embeddings generated by our method are of high quality as indicated by superior performance of our method. In particular, corrected embeddings generalize across different backbones (CLIP, SigLIP, ALIGN), different datasets beyond NegBench (CC-Neg, VALSE Existence, Flickr_30K subset). We further check the alignment between text embeddings with image embeddings in Table 2. and 3. The results show better alignment when using embedding corrections. We believe these results validate quality of generated embeddings. Reviewer may let us know if they have any specific concerns or want us to conduct any additional experiments.
>
> **Q2. Evaluation on MLLMs and application to LLaVA**
> This is a thoughtful question. We answer the reviewer's concern by first discussing the MLLM benchmark and then potential impact of our approach.
> 1. **MLLM benchmark:** This benchmark is used to evaluate performance of general multimodal large language models with generative capability. Focus of present work is to improve semantic alignment of joint embedding space features, such as CLIP text embeddings, so that these embeddings represent semantic information in the text more accurately by understanding the negated concepts. Since MLLM benchmarks do not represent texts with negation, potential benefits of using our approach will be limited.
> 2. **Application to LLaVA models:** We note that, our method corrects CLIP text encoder embeddings and multimodal LLMs such as LLaVA use CLIP Image encoders to understand Image data. Further, text is processed by high capacity LLM models such as LLAMA family which have much better language understanding due to large number of parameters (several Billions) and internet scale language pre-training. Thus direct application of our approach to LLaVA model family is incompatible as we address different problem (better image and text alignment with negation understanding) and multimodal LLM address different problem (text generation by analyzing text and visual data).

---

> > ### Author Response · Authors · 2025-11-22
> > **Authors’ Response to Reviewer YhKj**
> >
> > **Q4. Effect of synonyms and the quality and diversity of corrected embedding**
> > This is an interesting question and a good ablation to validate robustness of our approach. We believe the reviewers' concern is about the robustness of the corrected embeddings for captions with synonymous meanings. **We exactly perform this experiment and show the embeddings in Fig.6 in the paper**. In particular, Fig.6-right shows embeddings generated for complex captions with synonymous meanings. For example, "The cat is visible in this image.", "This is a photo of a cat." are synonymous captions with affirmation. Similarly, "There is no cat here.", and "The cat is absent in this photo." are synonymous captions with negation. From the figure, we can see that proposed approach clusters these embeddings into different semantic regions in the feature space. This show that proposed **corrected embeddings are diverse (under different semantic concepts) and high quality (clustered together for similar concepts)**.  We hope this clarifies reviewer's concern.
> >
> > **Q5. Effect on general retrieval, VQA, classification**
> > Our approach improves negation understanding of CLIP text encoder by explicitly correcting embeddings using proposed approach. When negation is absent from given text (general retrieval), the embeddings are not updated and are left as is (Alg.1). In general common image and text retrieval benchmarks do not have negations in the text captions and as a result our method will get same accuracy as the baseline model. As shown in Table. 1 in the paper, for COCO/MSRvtt retrieval tasks, our method achieves same accuracy as original model, and for retrieval with negations, it achieves the best accuracies improving CLIP by 5.9% and 5.5% and NegCLIP by 0.9% and 0.7%. In summary, for general tasks such as retrieval, classification and VQA when negation is not present, the accuracy of our method should be same as original method. For the tasks with negations our approach get significant boost in accuracy (on NegBench, CC-Neg, VALSE, etc.).

---

> ### Comment · Reviewer_YhKj · 2025-11-25
>
> Thank you for providing a comprehensive rebuttal. Actually, there exists some negation VQA benchmarks, for example, NegVQA: Can Vision Language Models Understand Negation? If your method has a wider application value in some important downstream fields, addressing some critical problem, the contribution of the manuscript will be better highlighted. Besides, the evaluation results on backbones like SigLip2 is very good. Do you have plans to release various variants of boosted SigLip2?

---

> ### Author Response · Authors · 2025-11-25
> **Authors’ Response to Reviewer YhKj**
>
> Thank you for your suggestion regarding evaluation on NegVQA.
>
> We contend that evaluating our method on NegVQA or general VQA tasks is beyond the scope of this work due to a fundamental architectural and task mismatch between our target models (Contrastive VLMs) and Multimodal LLMs (MLLMs).
>
> Our method is a simple, zero-shot, geometric correction designed to fix the negation bias in the static joint-embedding space of CLIP-style models.
>
> MLLMs already possess powerful, pre-trained LLMs with sophisticated negation understanding. Their VQA failures (including on NegVQA) are typically related to complex visual grounding or multi-step reasoning, not the simple representational flaw our method corrects. Integrating our static vector correction into their dynamic, cross-attention-heavy architecture is non-trivial and fundamentally alters the nature of our training-free approach.
>
> **Wider Application Value**
> We are confident that our contribution is already well-highlighted by its impact on the foundational downstream applications that rely directly on a correctly-formed joint CLIP embedding:
>
> Text-to-Image Generation (T2I): Correcting the CLIP text embedding is paramount for T2I diffusion models to respect negative prompts, allowing for accurate exclusion of objects (e.g., generating "A person walking without a bag" as shown in the Figure.1).
>
> Fine-Grained Retrieval & Filtering: Our method significantly improves the expressiveness of the text encoder for both image and video retrieval, enabling accurate search queries that explicitly exclude concepts (e.g., "A beach with no people").
>
> Caption Verification & Hallucination Detection: Our approach is ideal for verifying the integrity of generated text through counter-factual checks (e.g., validating a hallucinated caption by checking its negated counterpart), providing a robust tool for automatic caption filtering.
>
> These examples confirm that our approach effectively addresses a critical, low-level representational problem in the VLM domain, yielding significant and practical benefits for high-value downstream applications.
>
>
> Regarding releasing SigLip2:
> We will make our pipeline public so that research community can reproduce our results on different models and datasets.

---

> > ### Comment · Reviewer_YhKj · 2025-11-25
> >
> > You have mentioned the visual captioning task. Including some downstream applications like T2I generation, VQA, and captioning can better highlight the comprehensiveness of the submission. I have raised my rating to a positive one.

---

> > > ### Author Response · Authors · 2025-11-25
> > > **Authors’ Response to Reviewer YhKj**
> > >
> > > Dear Reviewer,
> > >
> > > Thank you for increasing the paper score. We really appreciate it.

---

### Official Review · Reviewer_u7Qg · 2025-11-01

**Soundness:** 2
**Presentation:** 3
**Contribution:** 2
**Rating:** 2
**Confidence:** 4

**Summary:**

his paper aims to improve negation understanding in embedding-based vision–language models such as CLIP, which struggle to correctly interpret negated textual inputs. Prior approaches have mainly relied on data-centric solutions that fine-tune models on curated negation datasets, but these tend to be resource-intensive and sometimes degrade general performance.

The authors propose a zero-shot, training-free approach that decomposes a given text query into affirmative and negated parts using a rule-based negation detection algorithm, then applies a linear correction in embedding space to form a corrected “negation-aware” text embedding. The method achieves strong performance improvements on NegBench, boosting CLIP’s retrieval accuracy from 50.9% to 56.1% and multiple-choice (MCQ) accuracy from 25.5% to 67.0%.

**Strengths:**

1- the proposed approach is efficient, zero-shot, and training-free, and easy to integrate into existing CLIP-based pipelines.

2- the paper shows consistent improvements on different model NegBench retrieval and MCQ tasks.

3- the paper is well organized and easy to follow.

**Weaknesses:**

1- the rule-based negation detection might overfit to the NegBench benchmark, which is synthetically generated using structured templates and paraphrasing; therefore, generalization to natural or free-form text is uncertain.

2- as the authors discussed in the limitation section, the method’s coverage of negation cues is incomplete. This approach misses implicit or compositional negations (e.g. “barely visible,” “few people”, "alone"), and may mis-handle nested clauses or double negation.

3- although the improvement on the MCQ task appears large (+50% relative), the absolute task structure (4 options) limits interpretability since a +25% absolute increase is effectively equivalent to eliminating one distractor option.

**Questions:**

**Some questions and suggestions**

1- have you considered replacing the rule-based negation scope detection with a learned or LLM-based decomposition approach? It might improve coverage and robustness to natural language variation.

2- the NLP community has proposed several deterministic negation detection methods (e.g., NegEx, DEEPEN, or syntactic-scope models) that go beyond keyword matching. Could you compare performance to some of those?

3- have you evaluated the approach on non-synthetic or human-written captions like real image descriptions to test out-of-distribution generalization?

---

> ### Author Response · Authors · 2025-11-22
> **Authors’ Response to Reviewer u7Qg**
>
> Dear Reviewer,
> We sincerely thank you for your thorough assessment and constructive feedback. We are happy that you recognized the core strengths of our work, particularly its efficiency, zero-shot, and training-free nature, and the consistent, strong performance improvement. Your critical questions concerning generalization and the limitations of the rule-based component are addressed below.
>
> **Addressing Weaknesses**
> **W1: Rule-based negation detection and concerns over generalization to free form text**
> We acknowledge the concern about potential overfitting to synthetic data; however, we emphasize that the success of our method lies in exploiting the **compositional linearity** inherent in the *pre-trained Vision-Language Model (VLM) embedding space*, a fundamental property, rather than just optimizing a rule-set for syntax.
> 1.  **Fundamental Mechanism:** Our core contribution is the **embedding correction paradigm**—using vector arithmetic to explicitly remove semantic information. This principle is model and domain-agnostic.
> 2.  **Generalization Superiority:** Beyond outperforming fine-tuned methods like NegCLIP* on NegBench, we have additionally evaluated our model on the CC-Neg (shown in Appendix) and VALSE (See the response below) datasets, as well as on different VLM backbones including SigLip 1 & 2 and ALIGN (See the response to G9hW), showing superior and consistent generalization across diverse models and linguistic benchmarks.
> 3.  **Preservation of General Performance:** Our zero-shot approach maintains competitive or superior performance on general zero-shot tasks (e.g., ImageNet), demonstrating that our method does not induce **catastrophic forgetting**, a common failure mode for fine-tuned negation methods.
>
> **W2: Incomplete coverage of negation cues (implicit, compositional) and mis-handling of nested/double negation**
> We agree that our current rule-based parser represents a trade-off, prioritizing a **zero-shot, training-free** solution.
>  1.  **Context of Benchmarks:** It is important to note that all concurrent methods, including the original benchmarks for negation understanding (NegBench, CC-Neg, NegRefCOCO, VALSE, etc.), primarily use simple explicit negations. The reviewer's comments highlight the limitations of existing VLM evaluation benchmarks, where only simple negation structures are predominantly used.
> 2.  **Core Contribution Reiteration:** We again highlight that our central contribution is the **embedding correction method**—the novel vector space arithmetic—not just the rule-based parsing itself. We compare our methodology fairly against state-of-the-art backbones and latest methods on existing benchmarks, and the modular nature of our method allows for future drop-in replacement of the new parsing component.
>
> **W3: The interpretability of the large MCQ improvement**
> We respectfully clarify both the statistical significance of our results and the validity of our evaluation protocol.
> 1.  **Statistical Interpretation:** The reviewer suggests that the performance gain is equivalent to eliminating a single distractor. However, mathematically, eliminating one option from four raises the random baseline from 25% only to $1/3 \approx 33.3\%$. Our model achieved $67.0\%$ accuracy (average across all MCQ tasks), representing an absolute improvement of **$42\%$** over the random baseline. This far exceeds the theoretical ceiling of eliminating even two distractors ($1/2 = 50\%$), confirming that the model is relying on strong visual-semantic reasoning rather than simple elimination heuristics.
> 2.  **Adherence to Community Benchmarks:** We utilized the 4-option MCQ format specifically to align with the established evaluation protocols used by state-of-the-art LLM and VLM benchmarks for reasoning tasks, such as **MMLU [1]**, **MMMU [2]**, and **ScienceQA [3]**. By adhering to this widely accepted standard, we ensure our results are directly comparable to the broader research landscape. The substantial absolute gain reported is meaningful within this standard framework.

---

> ### Author Response · Authors · 2025-11-22
> **Authors’ Response to Reviewer u7Qg**
>
> **Addressing Questions**
> **Q1: Regarding use of learned or LLM-based decomposition approach?**
> This is a crucial direction for future work.
> 1.  **Efficiency Constraint:** We deliberately avoided integrating large language models (LLMs) in our current zero-shot method because LLMs are **very compute-intensive**, which runs counter to the paper's core philosophy of providing an **efficient, low-latency, training-free** solution.
> 2.  **Future Directions:** We envision several ways to integrate learned components:
>     * An improved rule-based parser can be a **drop-in replacement** for a more robust sentence parsing model in future iterations.
>     * We can leverage LLMs to **generate high-quality negation data** with complex implicit/explicit negations, which can then be used to test OOD generalization.
>     * For training-based methods, we propose training a VLM itself (similar to NegCLIP or NegationCLIP) but enforce our **embedding correction as a regularizer** during training. This enforces the compositional properties identified by our work at the fundamental training level. And eliminates the need for rule based parser during testing.
>
> **Q2: Comparison with other negation detection methods (e.g., NegEx, DEEPEN)**
> Thank you for the suggestion. Our current rule-based approach is **significantly motivated by established clinical NLP tools like NegEx and DEEPEN**. The comparative evaluation on NegBenchCOCO for CLIP backbone using our custom VLM-focused rules and an off-the-shelf NegEx implementation are shown below:
> | Method | Acc |
> | :--- | :---: |
> | **Our** | **72.5** |
> | NegEx | 61.3 |
>
> As can be noted, NegEx based caption parsing resulted in lower  performance of 61.3% compared to ours 72.5. We note that we use NegEx simply as an off-the-shelf component here, to parse the negated concept from the caption. Upon detailed investigation, we note that, NegEx "termsets" does not include some of the common negations in natural language and as a result fails to parse many captions correctly leading to lower performance. Following are missing negation words which our rule based approach includes:
> **Missing Pre Negation Words:** 'lacks', 'excludes', 'won't', 'devoid of', 'absent'
> **Missing Post Negation Words:** 'is not', 'is nowhere', 'is absent'
> We believe that, once these negators are included in the NegEx ternsets, its performance will likely improve and may match with our final accuracy. However, we again note that, the primary contribution of our work is not the rule based parsing algorithm, but the proposed negation aware embedding correction method thus we did not focus more on improving NegEx.
>
> **Q3: OOD generalization and evaluation on non-synthetic or human-written captions like real image descriptions**
> This is a key test for robustness, and we address it below:
> 1.  **Existing OOD Evidence:** As noted in response to W1, we have evaluated our method on the CC-Neg, highlighting generalization across different benchmarks beyond the NegBench templates. We additionally evaluate our approach on VALSE Existence dataset with 2 option MCQ benchmark below:
> | Model | Acc |
> | :--- | :---: |
> | CLIP | 70.97 |
> | Con-CLIP | 71.72 |
> | NegationCLIP | 78.33 |
> | **CLIP+Ours** | **79.49 ($\uparrow$8.52)** |
>
> 2.  **Human Annotated Negations:** In general most of the modern datasets and benchmarks are created using carefully curated semi-automatic pipelines such as CC-Neg, NegBench, etc. We found Flickr30K_Negation [4] as the only dataset with human annotated captions with negation with 896 samples. As a quick sanity check for testing generalization of our embedding correction method, we created MCQ task on this dataset by combining random captions as multiple options. However, since images and captions are quite diverse, the original CLIP model achieves very high accuracy >97% on this task due to absence of hard negatives. Thus we increase the difficulty of this dataset by using all 896 available captions as MCQ options (this is similar to text retrieval task). We estimate the text retrieval accuracy for CLIP model this  way and report top 5 (R@5) and top 1 (R@1) accuracies.
> | Model | Acc R@5 | Acc R@1 |
> | :--- | :---: | :---: |
> | CLIP | 67.41% | 34.04% |
> | **CLIP+Ours** | **70.42%** | **41.29%** |
>
> These results show that our embedding correction method improves alignment between image and text embeddings and generalizes to human annotations as well improving text retrieval accuracy from 67.4% to 70.4% on top-5 retrieval and 34% to 41.3% for top-1 retrieval tasks. We are confident that our zero-shot vector arithmetic approach offers a powerful, resource-efficient, and generalizable solution to the negation problem in VLMs.

---

> > ### Author Response · Authors · 2025-11-22
> > **Authors’ Response to Reviewer u7Qg**
> >
> > References
> > 1. Yue, X., et al. (2024). "MMMU: A Massive Multi-discipline Multimodal Understanding and Reasoning Benchmark for Expert AGI." Proceedings of the IEEE/CVF Conference on Computer Vision and Pattern Recognition (CVPR).
> > 2. Hendrycks, D., et al. (2021). "Measuring Massive Multitask Language Understanding." International Conference on Learning Representations (ICLR).
> > 3. Lu, P., et al. (2022). "Learn to Explain: Multimodal Reasoning via Thought Chains for Science Question Answering." Advances in Neural Information Processing Systems (NeurIPS).
> > 4. Miltenburg, V., et al. (2016). "Pragmatic Factors in Image Description: The Case of Negations." Proceedings of the 5th Workshop on Vision and Language.

---

> ### Comment · Reviewer_u7Qg · 2025-11-22
>
> Thanks for the detailed response. I have a few follow-up comments:
>
> 1- On using rule-based splitting: it’s good that you added a comparison with NegEx to separate the positive and negative parts of the query. I also understand your choice of not using LLMs to extract segments for efficiency reasons “LLMs are very compute-intensive, which runs counter to the paper’s core philosophy of providing an efficient, low-latency, training-free solution”
> Given it's the case, I think it would help to include an additional column in your tables that uses an “Oracle/LLM” split. That would make the time–accuracy trade-offs clearer between your rule-based split, NegEx, and an ideal segmentation.
>
> 2- On synthetic vs real benchmarks: adding Flickr30K_Negation does address my earlier concern about overfitting to synthetic benchmarks such as NegBench. When a benchmark is generated by paraphrasing “A but not B”, splitting the query with a rule-based model can unintentionally become a form of reverse engineering or “hacking” the benchmark. This is why evaluating on real datasets—even if small-scale—is important.
> I think the paper should more clearly distinguish results on synthetic vs. real negation benchmarks.
>
> 3- I agree that the most interesting part is the embedding modification itself, rather than the idea of splitting the query. I would like to see some analysis of how the output of your approach differs from data-centric ones like the fine-tuned CLIP on NegBench. Given the timeline, some qualitative examples comparing your method to data-centric approaches would be enough for me (e.g., examples where the behavior diverges).
>
> Minor comment:
> For Figure 3, I only see 11 clusters in the CLIP embedding example while you mention 20 objects, and the plot looks different from the one in the original NegBench. Maybe the color scheme or cluster labeling needs to be adjusted?
>
> If the authors can address my comments above, I would be happy to raise my score. Also, just to double-check, have you updated the paper with these new results? It doesn’t seem to show the updated content on my end.

---

> > ### Author Response · Authors · 2025-11-25
> > **Authors’ Response to Reviewer u7Qg**
> >
> > Dear Reviewer,
> >
> > Thank you for your comment.
> >
> > 1. We have included results on human annotated benchmarks as a subsection within result section (Table 4) to highlight evaluation on synthetic vs real datasets more clearly.
> >
> >
> > 2. We have included some qualitative examples in the Appendix section showing failure modes of NegCLIP* model.
> >
> > 3. Comparison of rule based parser with LLM and NegEx: We added accuracy comparison between NegEx and our method in Appendix Table.10. We are currently working on parsing NegBench-COCO-MCQ dataset with llama models. Since correctly parsing captions with negation requires careful prompt design to get correct results and due to lack of enough GPU resources on our end, the process may take some time. We will update Table.10 to include LLM (Oracle) results when we complete the task.
> >
> > 4. Regarding Fig.3: As shown in the legend of the figure, we have included 16 objects. We have used **husl color palette** to show object samples in the plots. Unfortunately, we did not realize this color palette generated similar colors for different objects and hence the confusion.
> >
> > We have incorporated all the suggestions in the revised manuscript which is uploaded now. We are really happy to know the willingness of the reviewer to increase the paper score.
> >
> > Thank You.

---

> > > ### Comment · Reviewer_u7Qg · 2025-11-28
> > >
> > > Thanks for the updates. Since you’re adding the accuracy/performance trade-offs for the rule-based approach and LLMs, I will raise my score.
> > >
> > > Minor comment: Figure 3’s colors don't seem updated yet, and the color palette is missing in Figure 6.

---

> > > > ### Author Response · Authors · 2025-12-01
> > > > **Authors’ Response to Reviewer u7Qg**
> > > >
> > > > We really appreciate your willingness to raise the score!
> > > >
> > > > Regarding your suggestion to include an "Oracle/LLM" split for the negation parser comparison:
> > > > We attempted to generate these oracle results using **Llama models (llama 4 scout)**, using extensive prompt tuning with over 50 in-context learning examples. Unexpectedly, the LLM-parsed captions yielded only **64.7%** accuracy on NegBench-COCO-MCQ. While this outperforms NegEx (61.3%), it is **significantly lower** than our rule-based parser (72.5%).
> > > >
> > > > We believe that the LLM is either clearly struggling with instruction-following or is hallucinating (we found several cases where it failed to parse any negation at all). Although this issue is fixable with more time for exhaustive prompt tuning and additional GPU resources, we are constrained by the short rebuttal timeline.
> > > >
> > > > Given this, we unfortunately cannot guarantee the timely inclusion of reliable "Oracle" split results. We sincerely hope the existing comparison of our method (72.5%) versus NegEx (61.3%) serves as sufficient ablation for now.

---

### Official Review · Reviewer_G9hW · 2025-11-02

**Soundness:** 3
**Presentation:** 3
**Contribution:** 3
**Rating:** 8
**Confidence:** 4

**Summary:**

This paper proposes a training-free text-embedding edit for CLIP to handle negation: detect the negated span with rules, subtract the projection of the full caption embedding onto the negated-concept embedding, and add an “anchor” vector (Eq. 2). On NegBench, MCQ accuracy jumps from 24.7/24.3/27.5 (COCO/VOC/MSRVTT) to 72.5/78.6/50.0 for CLIP; negated-caption retrieval R-Neg@5 improves modestly (COCO +5.9; MSRVTT +5.5).

**Strengths:**

- Simple, training-free mechanism; clear formula and pipeline (Alg. 1, Eq. 2).

- Large MCQ gains on NegBench without changing positive retrieval (R@5).

- Geometry story (negation axis) supported by PCA visualizations (Fig. 3 and 4, Embedding space arithmetic visualization).

**Weaknesses:**

- Missing relevant related work: prior work [1] shows inference-time steering using CLIP geometry and visual-semantic arithmetic; it should be cited and contrasted (LM-side steering vs. text-side embedding edit).

- Generalization beyond CLIP text towers (e.g., SigLIP/ALIGN, BLIP text encoders, MLLMs) is untested.

[1] Tewel, Yoad, et al. "Zerocap: Zero-shot image-to-text generation for visual-semantic arithmetic." Proceedings of the IEEE/CVF conference on computer vision and pattern recognition. 2022.

**Questions:**

- It would be great to see if the proposed approach generalizes to similar but more recent models, such as SigLIP families. Is there any particular reason it was not tested on these?

---

> ### Author Response · Authors · 2025-11-22
> **Authors’ Response to Reviewer G9hW**
>
> We sincerely thank the reviewer for their thorough review, positive assessment, and valuable constructive feedback regarding related work and generalization. We address your points below:
>
> **1. Missing Relevant Related Work (Tewel et al. / Zerocap)**
>
> We acknowledge the oversight and agree that **Tewel et al. (2022), "Zerocap,"** is highly relevant and should be cited and contrasted. We will ensure this is included in the final manuscript. The key distinction between our work and Zerocap is as below:
> **Our approach:** We present a **zero-shot, training-free text-side embedding correction.* Our method directly manipulates the final **CLIP text embedding** through arithmetic operations ($e^* = e^c - \lambda * ( Proj_{e^c}(e^{neg} - a$) to enforce the effect of negation *before* the cross-modal comparison. It is an alteration of the semantic representation itself for better retrieval and classification.
> **Zerocap (Tewel et al., 2022):** It uses visual-semantic arithmetic for **LM-side steering** to enable *zero-shot image-to-text generation*. This is an inference-time modification within the sequence generation process of a language model (LLM), typically to control the *output* text based on visual cue arithmetic (addition or subtraction of image embeddings), which is a different goal and application domain.
>
> However, the fundamental common principle between both methods is **the zero-shot use of arithmetic operations to manipulate semantic information within the CLIP embedding space**. Our approach modifies text embeddings to correct negation bias in classification/retrieval, whereas one key application of Zerocap is the manipulation of image embeddings (addition and subtraction) to achieve compositional control for text generation. We will clarify this shared principle and the differing application domains in our Related Work section.
>
> **2. Generalization to Other Models (SigLIP)**
> We apologize for overlooking the implementation on other backbones. We have extended our evaluation to more recent and structurally similar models, including SigLIP 1 & 2, ALIGN, and TripletCLIP. We found that the core principle of our method—leveraging embedding arithmetic for negation correction—successfully generalizes across these models. Our approach yields similar performance boost on the NegBench benchmark for these architectures. This robust generalization strongly supports our central claim: the representational deficiency concerning negation is not unique to CLIP but is a systemic issue in large-scale joint visual-semantic models trained via contrastive objectives on web data. Our simple, training-free text-embedding edit provides an effective solution to this problem across diverse VLM families. We will include a dedicated table and discussion of these results in the revised manuscript. We use $\lambda$ 1.9 for all experiments.
>
> | Models | COCO | VOC | MSRvtt | COCO Ret R@5 | COCO Ret R-Neg@5 | MSRvtt Ret R@5 | MSRvtt Ret R-Neg@5 |
> | :--- | :---: | :---: | :---: | :---: | :---: | :---: | :---: |
> | SigLIP | 28.9 | 30.8 | 30.7 | 72.1 | 64.4 | 51.4 | 44.7 |
> | **SigLIP+Ours** | **61.1 ($\uparrow$32.2)** | **66.9 ($\uparrow$36.1)** | **46.3 ($\uparrow$15.6)** | - | **68.7 ($\uparrow$4.2)** | - | **47.6 ($\uparrow$2.9)** |
> | SigLIP2 | 27.2 | 27.1 | 30.1 | 72.8 | 64.0 | 51.6 | 45.3 |
> | **SigLIP2+Ours** | **65.4 ($\uparrow$38.2)** | **68.5 ($\uparrow$41.4)** | **49.5 ($\uparrow$19.4)** | - | **70.2 ($\uparrow$6.2)** | - | **49.4 ($\uparrow$4.1)** |
> | ALIGN CLIP | 32.7 | 28.4 | 23.6 | 44.8 | 35.6 | 35.8 | 31.1 |
> | **ALIGN CLIP+Ours** | **59.4 ($\uparrow$26.7)** | **68.4 ($\uparrow$40.0)** | **43.8 ($\uparrow$20.2)** | - | **41.2 ($\uparrow$5.6)** | - | **34.7 ($\uparrow$3.6)** |
> | TripletCLIP | 33.8 | 23.7 | 30.2 | 52.3 | 44.3 | 42.4 | 35.8 |
> | **TripletCLIP+Ours** | **61.8 ($\uparrow$28.0)** | **57.2 ($\uparrow$33.5)** | **46.7 ($\uparrow$16.5)** | - | **47.9 ($\uparrow$3.6)** | - | **39.9 ($\uparrow$4.1)** |

---

> > ### Comment · Reviewer_G9hW · 2025-11-26
> >
> > Thank you for addressing all questions! I think all revisions significantly improved the submitted work, and I'm keeping my positive score.

---

### Author Response · Authors · 2025-11-25
**General response from the Authors**

We once again thank all the reviewers for their time and comments. After incorporating their suggestions in the paper the quality of the revised manuscript improved significantly. Below, we provide a comprehensive summary of the changes.

1. Line078-094: We have rephrased sections of Introduction to reflect the motivation derived from prior works and our own experiments accommodating the comments from Reviewer aYJg.
2. We have incorporated table showing results on different backbones (Table.2) and on VALSE Existence (Table.3) and Flickr30K (Table.4) datasets as most of the reviewers requested additional evaluations.
3. We rephrased the Limitations section (Section 5. ) to reflect discussion on limitations of rule-based parser.
4. We have included comparison between our negation detection algorithm and NegEx in appendix (Table.10) as requested by Reviewer u7Qg and aYJg.
5. We have added few qualitative examples showing failure modes of fine tuned NegCLIP* model as requested by Reviewer u7Qg.

The main paper now spans 10 pages.

Thank You.

---

> ### Author Response · Authors · 2025-12-01
> **General response from the Authors**
>
> Latest Revision Details:
>
> We have revised results in Table 2. Previously we had used a single value $\lambda=1.9$ for all the models due to time constraints. Now we have tuned  $\lambda$ for each model separately using 10% split of NegBench-COCO-MCQ. This resulted in a slight improvements to SigLIP2 and AlignCLIP model results.

---

### Author Response · Authors · 2025-12-02
**General Response to Area Chairs, Senior Area Chairs and Program Chairs**

In the light of the recent OpenReview bug, which affected the ICLR review process, the ICLR PCs, unfortunately, have decided to roll back the updated paper scores to the pre-rebuttal period.
Thus we feel it necessary, to summarize the discussion between authors and reviewers.

**General Comments from Reviewers:**
The paper received overall positive score (5.8) and most of the reviewers had positive impression of the paper (8,8,6 rating by 3 reviewers out of 5). Reviewers raised minor concerns over additional validation on models, datasets and some clarification.

**Authors Response:**
Authors provided additional results (Table.2/3/4/10/15) and detailed clarification showing robustness, generalization and superior performance of method which **all reviewers agreed!**

**Summary:**
In conclusion,
- high scoring reveiwers G9hW(8) and aYJg(8) kept their score as is,
- positive scoring reviewer dika(6) increased their score further,
- low scoring reviewers YhKj(4) had increased score to 6 (now it is rolled back to 6) and u7Qg (2) also promised to increased their score to postive one!!!.

After a thorough and rigorous discussion during the rebuttal phase, **the authors were able to address all the concerns and all the reviewers agreed with rebuttals** and significantly improved their scores.

We hope that the Area Chairs will take this positive shift into consideration when making the final decision.

---

### Public Comment · ~DoYoung_Kim5 · 2026-05-26
**Plans for Code Release?**

Thank you for this interesting work.
I was wondering if there are any plans to release the source code to facilitate reproducibility and further research. It would be greatly appreciated by the community.

---

### Meta-Review · Area_Chair_6xBU · 2026-01-06

**Summary:**

This paper presents an approach for modeling negating in embedding-based VLMs. The approach involves subtracting the negated concept from the embedding. The approach leads to significant improvements on various benchmarks designed to test for negation understanding in embedding-based VLMs.

The reviewers generally agreed that the paper is well-written (although there seem to be issues with regard to type setting, as well as references), simple, and led to meaningfully improved results. On the negative side, most reviewers noted that the approach should be tested on other models, and that the use of rules for detecting negations was likely too limited (and tailored towards NegBench). Further, it is unclear how such an approach would generalize to non-embedding models.

I really really wish that the authors tried using an LLM to extract more sophisticated negations that cannot be extracted with simple rules, as this would have made contribution much stronger. In fact I encourage the authors to try to do this for the final version (even qualitative examples would be great). But despite this nontrivial limitation, I think the paper is above the acceptance bar.

On another note, this type of technique is related to "concept erasure" in NLP (e.g., https://aclanthology.org/2022.emnlp-main.405.pdf). I recommend the authors review the literature around this and discussion their contribution in this context.

**Reviewer Concerns:**

Major concerns by reviewers included:
- Generalization to other models (the initial approach was only tested on CLIP)
- The use of rules-based approach for detecting negation (which is only possible for benchmarks like NegBench) means that the approach won't generalize to more subtle negations.

The first point was addressed in the rebuttal, which led to several reviewers changing (or agreeing to change) their scores. The second point, which is quite significant, remain unaddressed.

**Reviewer Scores:**

Reviewer u7Qg would have changed their score given the rebuttal (explicitly stated in the discussion).
Reviewer G9hW stated that they would maintain their score.
Reviewer dika stated that they would raise their score.

---

### Decision · Program_Chairs · 2026-01-26

Accept (Poster)